# Reversible DNA condensation drives natural transformation

Joshua I. Santiago[1], Ishtiyaq Ahmed[2], Jeanette Hahn[2,3], Abigail Rubino[1], Heonhwa Choi[1], Guy Adami [1], David Dubnau [2,3] ✉, Matthew B. Neiditch [2] ✉ & Keith J. Mickolajczyk [1] ✉

Natural transformation drives the spread of antibiotic resistance among bacteria. The DNA receptor ComEA is essential for transporting external transforming DNA into the periplasm by an unknown mechanism. Here, single-molecule optical tweezers and electron microscopy approaches show that *Geobacillus stearothermophilus* ComEA forms dynamic oligomers on DNA that can switch between two conformations depending on local concentration. When ComEA sparsely decorates DNA, it forms bridging oligomers that condense the DNA to generate sub-pN pulling forces. When ComEA more fully decorates DNA, it forms non-bridging oligomers that decondense DNA and cannot generate force. Mutating ComEA to favor either bridging or non-bridging conformations causes transformation deficiency in *Bacillus subtilis*, meaning condensation and decondensation each play mechanistic roles. Our results show that ComEA reversibly condenses DNA during natural transformation, first producing force to pull DNA into the periplasm and then abating force production to promote transport into the cytoplasm.

Natural competence for transformation is a major mechanism of horizontal gene transfer in bacteria, central to population genome maintenance and to the spread of genes conferring competitive advantages such as antibiotic resistance[1-4]. While the catalog of bacterial proteins involved in natural transformation is known, the mechanisms by which they coordinate their activities to internalize and integrate transforming DNA (tDNA) remain poorly understood. In particular, the molecular details of how the transformation machinery generates the forces required to pull large, highly charged DNA molecules through both a cell wall and a cell membrane remain elusive. Answering these fundamental questions is essential for understanding horizontal gene transfer and the resulting spread of antibiotic resistance and vaccine escape that threaten human health[5].

Natural transformation includes two successive subprocesses: uptake and transport. In DNA uptake, tDNA is pulled out of the environment and across the thick peptidoglycan cell wall (Gram-positive species) or thin peptidoglycan cell wall plus outer membrane (Gram-negative species) and into the periplasm (Fig. 1a). During uptake, a transformation pilus interacts with a small section of tDNA and retracts, bringing this section of DNA into the periplasm. Next, the DNA receptor ComEA binds to tDNA, securing its attachment to the cell[1]. During transport, a single strand of tDNA is pulled into the cytoplasm, with the concomitant degradation of the non-transforming strand [6-8] (Fig. 1a). The channel protein ComEC and the ATP-dependent translocase ComFA, with its binding partner, ComFC, mediate these events[1]. Here, we focus on DNA uptake, where a gap in understanding exists between initial capture and complete uptake of tDNA into the periplasm.

The protein ComEA, initially discovered in *Bacillus subtilis* as a sequence-independent tDNA receptor, is known to play an essential role in tDNA uptake[9,10]. In Gram-positive bacteria, ComEA is a single-pass transmembrane protein with most of its mass in the periplasm

[1]Department of Biochemistry and Molecular Biology, Robert Wood Johnson Medical School, Rutgers University, Piscataway, NJ, USA. [2]Department of Microbiology, Biochemistry, and Molecular Genetics, New Jersey Medical School, Rutgers Biomedical Health Sciences, Newark, NJ, USA. [3]Public Health Research Institute, Rutgers Biomedical Health Sciences, Newark, NJ, USA. ✉e-mail: dubnauda@rutgers.edu; neiditmb@njms.rutgers.edu; keith.mickolajczyk@rutgers.edu

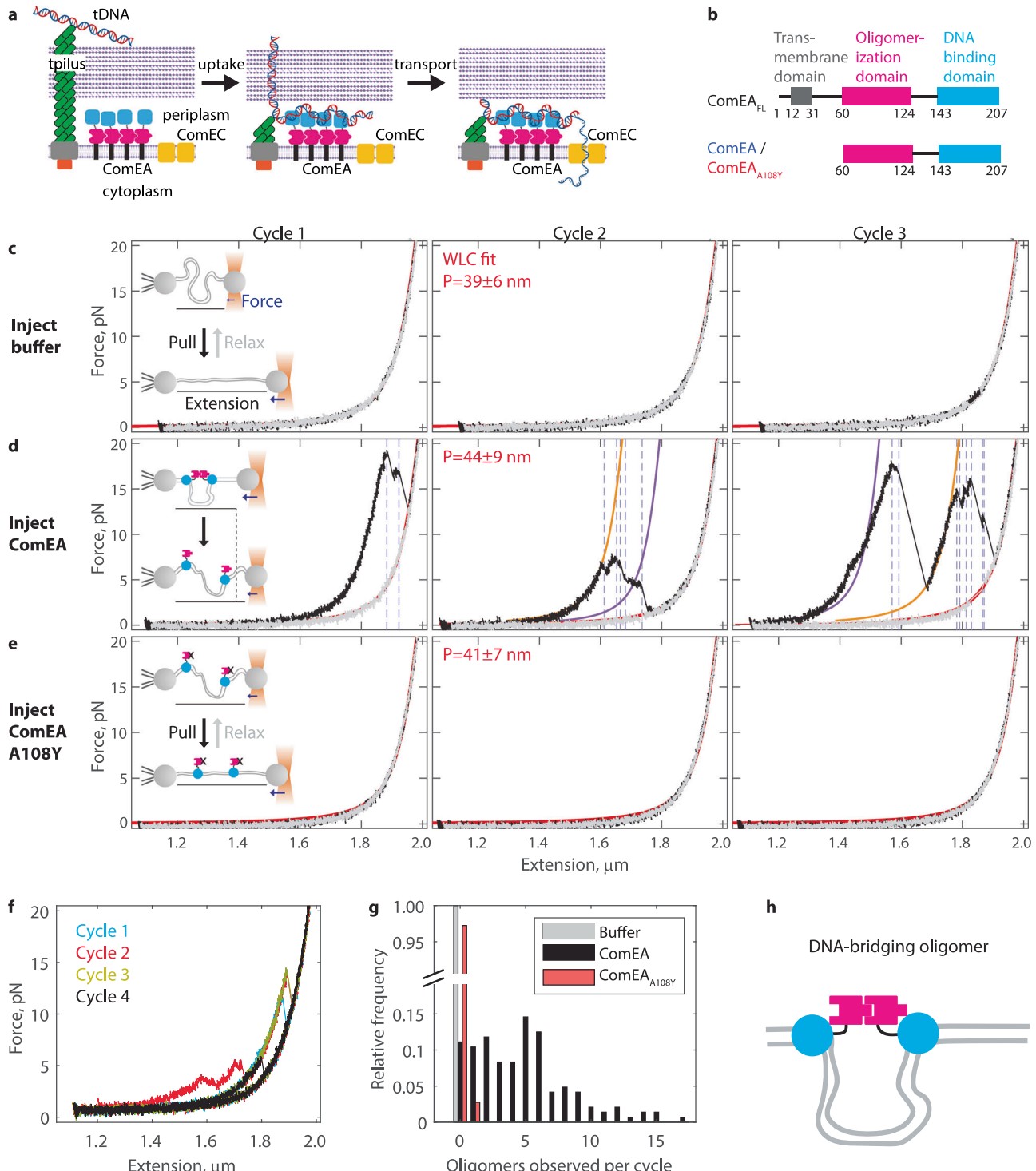

(Fig. 1a, b). In Gram-negative bacteria, ComEA comprises only a DNA-binding domain and freely diffuses in the periplasm[11–13]. Early models of tDNA uptake described the role of ComEA as a Brownian ratchet; after pilus retraction, additional lengths of tDNA can diffuse into the periplasm, while ComEA binding prevents backward diffusion[3,12,14,15]. However, our recent work showed that simple DNA binding is insufficient to explain the role of ComEA in DNA uptake[16]. We determined the high-resolution structure of recombinant ComEA from Gram-positive bacteria (Fig. 1b) and found that it contained a second, folded domain. Using analytical ultracentrifugation and other biochemical analyses, we showed that this domain enables oligomerization between ComEA molecules. We designed a structure-guided mutant, ComEA$_{A108Y}$

(Fig. 1b), that prevented self-assembly in biochemical assays. Critically, replacing ComEA with this oligomerization-deficient ComEA$_{A108Y}$ in vivo led to a nearly complete loss of transformability[16]. Hence, the oligomerization of ComEA within the periplasm is essential for successful tDNA uptake, negating the simple DNA-binding based Brownian ratchet model for Gram-positive bacteria. Although the role of ComEA oligomerization in tDNA uptake remained obscure, it was suggested that it may provide a pulling force to drag tDNA into the periplasm.

The current work tests and extends this idea, using reconstitution biochemistry and single-molecule optical tweezers to investigate the hypothesis that ComEA oligomerization can actively pull DNA inwards.

**Fig. 1 | ComEA forms DNA-bridging oligomers. a** Diagram of natural transformation in Gram-positive bacteria, indicating DNA uptake and transport steps. The pilus is comprised of ComGC (green), ComGB (gray), and ComGA (orange). ComEA (pink/blue) binds DNA in the periplasm. **b** Domain diagrams for recombinant ComEA from *Geobacillus stearothermophilus*. Constructs used in this work (amino acids 60-207) include the oligomerization domain and DNA binding domain. A structure-guided point mutation, ComEA$_{A108Y}$, renders ComEA oligomerization deficient. **c** Example force extension data for 6 kbp dsDNA after injecting buffer into the flow chamber. Three successive cycles of pulling (black) and relaxing (gray) are shown. Fits to the extensible worm-like chain equation are shown in red. Fitted persistence length for $n = 105$ cycles from $N = 13$ DNA molecules shown in the inset (mean ± standard deviation, SD). **d** Example force extension data after injection of 200 nM ComEA into the chamber. Oligomer breakage is observed as immediate changes to longer extension/lower force, denoted here by dotted blue lines. Sections before oligomer breakage were fitted to the eWLC equation; fits that matched the data well are shown in orange, while fits that matched poorly are shown in purple. Fitted persistence lengths for the relaxation phase for $n = 144$ cycles from $N = 14$ DNA molecules are shown in the inset (mean ± SD). **e** Example force extension data after injecting 200 nM ComEA$_{A108Y}$, a mutant generated to enable DNA binding but disrupt oligomerization. Fitted persistence length for $n = 145$ cycles from $N = 14$ DNA molecules is shown in the inset (mean ± SD). **f** Four successive force extension curves in the presence of 200 nM ComEA, wherein an oligomer is observed forming in the same position repeatedly. **g** Histogram of the number of oligomers observed per cycle for conditions where buffer ($n = 105$ cycles from $N = 13$ DNA molecules), 200 nM ComEA ($n = 144$ cycles from $N = 14$ DNA molecules), or 200 nM ComEA$_{A108Y}$ ($n = 145$ cycles from $N = 14$ DNA molecules) was injected. **h** Cartoon of a DNA-bridging dimer. The DNA-binding domains of ComEA (cyan) are at distal positions along the DNA (gray), but the oligomerization domains (pink) are connected. This conformation sequesters a loop of DNA and leads to DNA condensation. Elements of this figure were created in BioRender. Neiditch, M. (2026) [https://BioRender.com/uq5flkp].

Analyzing the dynamic interactions of ComEA pairs bound to DNA, we find that oligomerization is sufficient to condense DNA and produce pulling forces in the sub-pN regime. Excitingly, increasing the number of ComEA molecules per DNA molecule leads to a conformational change in the organization of ComEA oligomers that turns off force production and decondenses DNA. This concentration-dependent molecular switch, in which ComEA forms force-generating types of oligomers at low concentration to condense DNA and passive types of oligomers at high concentration to decondense DNA, underlies the handoff between DNA uptake (pulling tDNA into the periplasm from the environment) and DNA transport (pulling tDNA into the cytoplasm from the periplasm). We generate separation of function mutants of ComEA that either fail to condense or fail to decondense DNA and find that each lead to natural transformation deficiency in vivo. Our results show that ComEA reversibly condenses DNA. At lower concentrations, ComEA produces force, condenses DNA, and drives its uptake into the periplasm. At higher concentrations, ComEA turns off force generation, decondenses DNA, and promotes its transport to the cytoplasm.

## Results

### ComEA drives DNA condensation by forming bridging oligomers

To directly probe the dynamics of ComEA oligomerization, we measured protein-induced changes to DNA structure using single-molecule optical tweezers and recombinant ComEA (Fig. 1b). The ComEA was a soluble truncation of ComEA from *Geobacillus stearothermophilus* lacking the N-terminal sequence, hereafter referred to as ComEA as opposed to ComEA$_{FL}$ [16]. A 6 kilobase pair (kbp) strand of double-stranded DNA (dsDNA) with a biotin on one 5' end and an azide group on the other was bound to two ~2.1 μm latex beads. One bead was held steady by suction in a micropipette, while the other was held in a steerable optical trap. Force extension cycles were then run by (1) holding the DNA at ~55% extension relative to the contour length for 10 s, (2) moving the trap forward at 100 nm/s until the DNA was near full extension (50 pN), and (3) moving the trap backwards at 100 nm/s to the resting extension (Fig. S1). After an individual DNA strand was identified, we injected a buffer blank (Fig. 1c), 200 nM ComEA (Fig. 1d), or 200 nM of oligomerization-deficient mutant ComEA$_{A108Y}$ (Fig. 1e) and ran successive force-extension cycles.

For the buffer blank (Fig. 1c), the pulling phase (black) and the relaxing phase (gray) of each cycle overlapped. The data were fit well with the extensible worm-like chain (eWLC) model (red lines) with parameters matching previously reported values for dsDNA [17–20]. We observed major differences after injecting ComEA (Fig. 1d; additional examples in Fig. S2). An increase in force was measured at shorter extensions during the pulling phase (black) and was followed by an instantaneous drop in force paired with a sudden burst forward in extension (events at dotted blue lines). Several such events were observed in each cycle, different each time, but similar in that after some number of drops, the pulling phase realigned with the relaxing phase. In many instances, the force-extension curve between events (blue dotted lines) fit well to the eWLC equation under the simple assumption that only the contour length could change (Fig. 1d orange curves). Given the short (7 bp) binding footprint of ComEA [16], these events are best understood as ComEA oligomers forming loops of DNA (Fig. 1d diagram inset), effectively shortening the DNA until the applied force breaks the oligomer, and the length of the loop is released. In other instances, the force-extension curve between events was unsmooth or moving backwards, such that it could not be fitted by the eWLC equation (Fig. 1d purple curves); potential explanations include the breakdown of higher-order oligomers or ComEA sliding under force prior to breaking and releasing the DNA loop. Given the possibility of forming and breaking the same oligomer in several successive cycles (Fig. 1f), ComEA is unlikely to freely diffuse along DNA or to detach after oligomer breakage. Importantly, events were almost never detected after injecting the oligomerization-deficient mutant ComEA$_{A108Y}$, nor were there changes to the DNA contour or persistence length, reinforcing that events with ComEA present are due to oligomerization (Fig. 1e, g). Overall, we conclude that ComEA can form DNA-bridging oligomers, where the non-diffusing DNA-binding domains of two or more molecules are bound at distal positions along the DNA, effectively condensing the DNA (Fig. 1h).

### ComEA-driven DNA condensation is partial

We next investigated the force-extension curves with 200 nM ComEA in quantitative detail. Although the subsections of the curve between oligomer breaking events could not be fit to the eWLC model in every instance, most commonly due to the subsection being very short, we could determine the datapoint before each breakage and compare it to a point of equivalent force on the subsequent relaxation curve (Fig. 2a). This style of analysis is agnostic to activity before breakage events [21–25], although we note that it only determines the amount of DNA released upon ComEA oligomer breakage, which may be smaller than the DNA loop size initially formed. The equation for DNA released is derived from the eWLC equation:

$$DNA\,released = \left(2.94\,\frac{bp}{nm}\right)\frac{\triangle x(F)}{\left(1 - \frac{1}{2}\left(\frac{k_B T}{PF}\right)^{1/2} + \frac{F}{S}\right)} \qquad (1)$$

Where $\triangle x$ is the distance in nm between the breakage point and the relaxation curve (Fig. 2a), F is the force at which breakage occurred, $P$ is the DNA persistence length (45 nm), and $S$ is the DNA stretch modulus (1200 pN; see "Methods"). Using this equation, the total DNA released

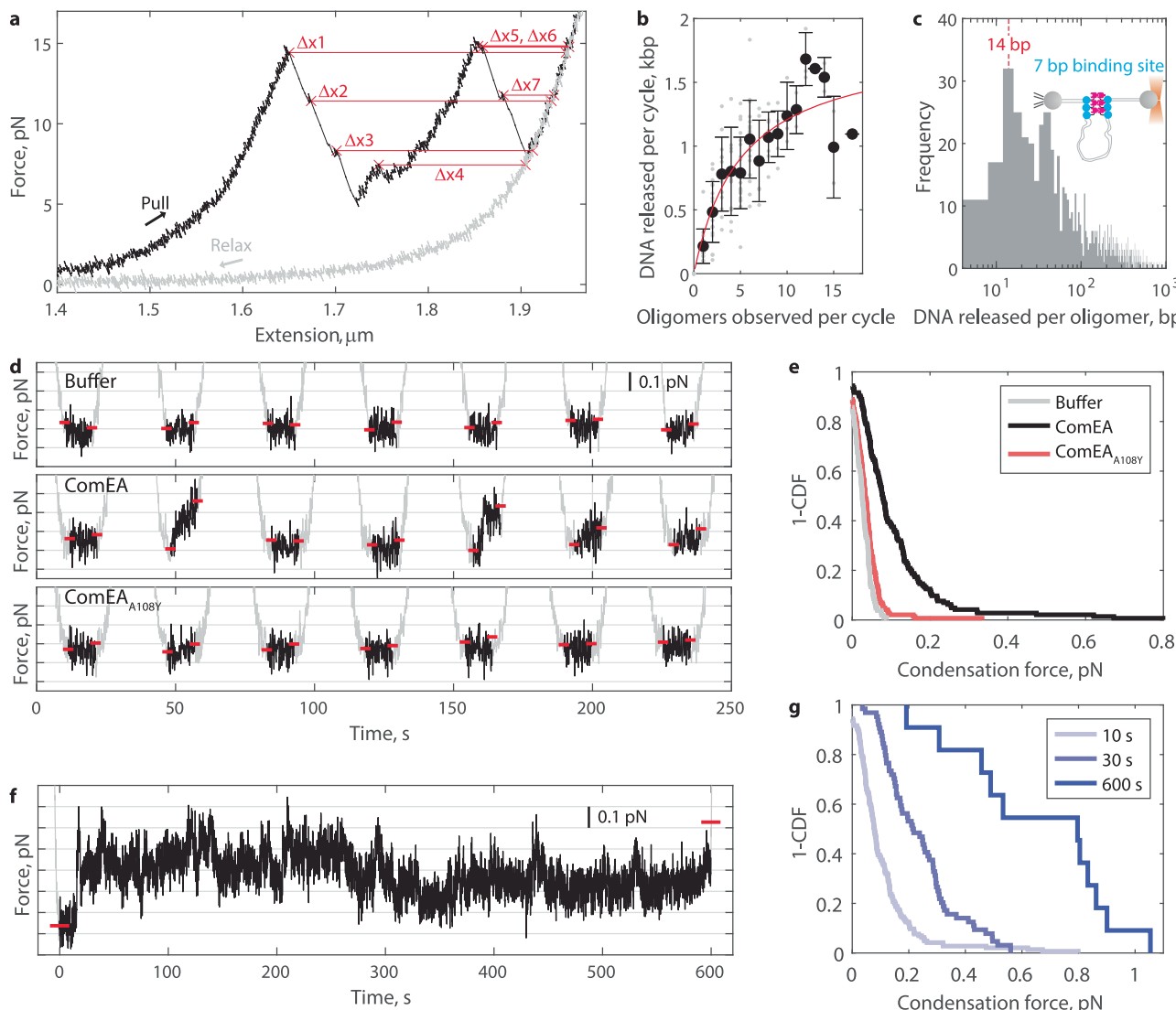

**Fig. 2 | ComEA oligomerization generates sub-pN DNA condensation forces.**
**a** Example force-extension curve with 200 nM ComEA present. Measuring the change in DNA extension ($\Delta x$) between the point before an oligomer breaks to that on the subsequent relaxation curve, at an equivalent force, enables calculation of the amount of DNA released by oligomer breakage. Each data point represents 5 ms. **b** The total amount of DNA released by oligomer breakage per cycle ($\Delta x1$ in **a**) plotted against the number of oligomers observed in that cycle ($n = 144$ total cycles). Individual measurements are shown as gray points, and the mean and standard deviations are shown in black. Fitted hyperbola in red, with EC50 = $5.1 \pm 4.7$ and $Y_{max}$=$1.82 \pm 0.62$ kbp, fit $\pm$ 95% confidence intervals (CI). **c** Histogram of the amount of DNA released by each observed oligomer breaking event ($N = 671$ events). **d** Example force versus time data during the 10-s rest phase of seven successive force-extension cycles after injecting buffer (top), 200 nM ComEA (middle), or 200 nM ComEA$_{A108Y}$ (bottom). The rest phases are colored black, while the pulling and relaxation phases are colored gray. The force axis is zoomed in and

truncated to highlight force production while the optical trap remains stationary. Horizontal red dashes denote the average force during the first second of rest (left side) and the highest force held for one second during the rest phase (right side). **e** Distributions, shown as survival plots (1 - cumulative density function, CDF), of the condensation force (i.e., the difference between the two red dashes in each cycle in **d**) generated during the 10 s rest phase of force extension curves in the presence of buffer alone ($n = 105$ cycles from $N = 13$ DNA molecules), 200 nM ComEA ($n = 144$ cycles from $N = 14$ DNA molecules), or 200 nM ComEA$_{A108Y}$ ($n = 145$ cycles from $N = 14$ DNA molecules). **f** Example force versus time data during a 600-s rest phase during a single force extension cycle after injecting 200 nM ComEA. The color coding matches **d**. **g** Distributions, shown as survival plots (1-CDF), of the condensation force generated by 200 nM ComEA during force-extension cycles where the rest phase was held for 10 s ($n = 144$ cycles from $N = 14$ DNA molecules), 30 s ($n = 64$ cycles from $N = 13$ DNA molecules), or 600 s ($n = 11$ cycles from $N = 5$ DNA molecules).

per cycle is determined using the first breakage ($\Delta x1$), and the DNA released by each oligomer is determined by subtracting the values measured for successive breakages ($\Delta x1$ and $\Delta x2$, for example).

The amount of DNA released per cycle was measured for $n = 144$ cycles and binned based on the number of oligomers observed in that given cycle. We expected to find that more oligomers would lead to more DNA sequestered into loops. Surprisingly, these data appeared hyperbolic, with the amount of DNA released per cycle plateauing at $1.82 \pm 0.62$ kbp, a value much lower than the 6 kbp DNA length and the ~3 kbp of DNA freely available for capture during the low-force resting

phase of the force-extension cycle (Fig. 2b). Hence, DNA-bridging oligomers of ComEA can only partially condense DNA. The amount of DNA released per oligomer was broadly distributed, with sizes ranging three orders of magnitude, but with a bias towards smaller releases (>50% smaller than 100 bp; Fig. 2c, linear scaling in Fig. S3a). The mode was centered at 14 bp. While it is possible that loops this small may form, we favor an interpretation where ComEA, which has a 7-bp binding footprint[16], may line up at the base of a loop, in which case each breakage closest to the base would release two binding footprints worth of DNA (Fig. 2c diagram inset). Consistently, 14 bp releases were

most commonly followed by another 14 bp release (Fig. S3a, b). Further experimental support for this interpretation is discussed below. Regardless, the prevalence of small release sizes indicates that DNA-bridging oligomers rarely sequester large DNA loops. Taken together, our data indicate that as ComEA forms more and more bridging oligomers on DNA, it tends to reinforce loops that have already been sequestered rather than sequestering new loops, leading to only partially condensed DNA.

## ComEA generates sub-piconewton DNA condensation forces

To further investigate the limited condensation driven by DNA-bridging ComEA oligomers, we analyzed the rest phases of the force-extension cycles. Here, the optical trap was stationary for 10 s, and the DNA was in a relaxed state (~55% extension; <0.1 pN of tension applied by the trap) such that DNA-bound ComEA may lock in DNA fluctuations via bridging oligomerization. If such oligomers formed a loop, the DNA inside the loop would be removed from the path of force transmission, effectively reducing the measurable DNA contour length. At a constant trap position, this will lead to the buildup of force. We measured the highest force produced and held for at least 1 s during each rest phase (see "Methods") after injecting buffer, 200 nM ComEA, or 200 nM ComEA$_{A108Y}$ (Fig. 2d). ComEA produced $0.11 \pm 0.01$ pN (mean ± SEM, n = 144), whereas buffer and ComEA$_{A108Y}$ lead to $0.03 \pm 0.01$ pN ($n = 105$) and $0.04 \pm 0.01$ pN ($n = 145$), respectively (Fig. 2e). As expected, the measured condensation force during a given rest phase scaled with the measured DNA released per cycle during the pulling phase of that cycle (Fig. S3c). These results show that DNA-bridging ComEA oligomerization generates measurable condensation forces.

We next increased the duration of the rest phase to see if ComEA could generate larger forces via bridging oligomerization if given sufficient time to lock in rare DNA fluctuations. Even when given 10 min, ComEA generated only a small amount of force; an equilibrium was quickly reached where bursts to higher or lower forces were readily returned to baseline (Fig. 2f). Such dynamics, detected at all rest phase durations, indicate that DNA-bridging oligomers both form and break when the optical trap is not moving (Fig. S4). Overall, the maximum force generated and held for at least 1 s stayed in the sub-pN regime for all rest phase durations tested (Fig. 2g).

As an alternate method for measuring ComEA oligomerization forces, we varied how far back from full contour length the DNA was held during the rest phase of each force-extension cycle (Fig. S5a). DNA-bridging oligomers were very rarely observed when the DNA was >75% extended during the rest phase but increased to >12 oligomers observed per cycle when the DNA was ~37% extended during the rest phase (Fig. S5b, c). Accordingly, the amount of DNA released per cycle increased nonlinearly (Fig. S5d). More than 1.82 kbp (Fig. 2b) could be sequestered/released if more DNA was exposed in the first place, but a plateau was always reached when the formation of another loop would require a particular condensation force to be generated. Fitting our data to an eWLC model, we determined the force to be $0.07 \pm 0.02$ pN, consistent with the direct force measurements above (Fig. 2e). Altogether, these results show that DNA-bridging ComEA oligomers can condense DNA until tension in the 0.1–1 pN range is reached, at which point new bridges fail to net form and condensation stalls. The fact that ComEA oligomerization can generate condensation forces is significant in explaining its essentiality for DNA uptake.

## Condensation by DNA-bridging ComEA oligomers is concentration-dependent

We next sought to directly visualize ComEA-driven DNA condensation using negative stain electron microscopy. A linearized 5.495 kbp dsDNA plasmid imaged alone appeared as a thin, wavy line, as expected for a polymer with short persistence length (Fig. 3a). Incubating the DNA with 200 nM ComEA led to the appearance of DNA loops (Fig. 3b),

consistent with the optical tweezers force-extension data showing DNA release upon oligomer breakage (Figs. 1 and 2). Indeed, the loop structures appeared reinforced at the base, as expected if several ComEA oligomers lined up there (Fig. 3b, yellow arrows; more examples in Fig. S6). This result is consistent with the prominent 14 bp step size of DNA released per oligomer in the optical tweezers data (Fig. 2c). Increasing the amount of ComEA incubated with the DNA from 200 to 500 nM led to more DNA loops and further partial condensation of the DNA (Fig. 3c). We thus expected to find that adding even more ComEA would fully condense the DNA. Surprisingly, further increasing the ComEA concentration to 2 or 4 μM did not lead to even more loops (Figs. 3d and S6). Instead, the loops disappeared progressively and the DNA decondensed, appearing similar to the condition where no ComEA was present. DNA loops were not observed with any concentration of ComEA$_{A108Y}$ added, confirming that the loops are formed by ComEA oligomerization (Fig. 3e–g; more examples in Fig. S6). We measured the fold-condensation of the DNA by comparing the molecular radius of the smallest circle that could be drawn to encompass the DNA, both with and without the addition of ComEA. These data produced a chevron-shaped plot peaking at ~3.5-fold condensation (Fig. 3h). This limited degree of condensation is consistent with the limited DNA released per cycle in the optical tweezers experiment (Fig. 2b). We confirmed that ComEA was bound to the DNA at high concentrations by measuring the DNA thickness at nonoverlapping regions; incubation with 4 μM ComEA or ComEA$_{A108Y}$ led to DNA that was significantly thicker than the no protein added condition, but not from each other (Fig. 3i). This result indicates that ComEA binds to DNA at high concentrations, but does not strongly impact thermally-driven fluctuations of DNA bending. Hence, ComEA oligomerization drives DNA condensation at lower but not higher concentrations, even though ComEA is bound to the DNA at those higher concentrations.

To support the negative stain electron microscopy data, we repeated the optical tweezers DNA force-extension experiments using a wide range of ComEA concentrations. We observed a chevron-shaped plot in this experiment as well; the number of oligomers observed per cycle increased between 20 and 150 nM ComEA, and then decreased at higher concentrations (Fig. S7a–e). Importantly, we also observed a chevron shape in the condensation force generated by ComEA, with a peak of 0.11 pN at 150 nM ComEA and monotonically decreasing values at higher concentrations (Fig. S7f). The amount of force measured with 4 μM ComEA present ($0.04 \pm 0.01$ pN, $n = 90$) was within error of the amount of force measured in buffer alone ($0.03 \pm 0.01$ pN, $n = 105$; Fig. 2e). Sequentially injecting higher concentrations of ComEA onto a single stand of DNA (e.g., 4 μM after 150 nM) led to changeover in behavior to the higher concentration faster than could be experimentally measured. Altogether, our results show that at lower concentrations ComEA can form bridging oligomers that produce force and condense the DNA, while at higher concentration ComEA passively binds to DNA and does not produce force or induce condensation.

## ComEA forms non-bridging oligomers at high concentrations

Our optical tweezers and electron microscopy data suggest that at high concentration, ComEA binds to DNA but does not induce condensation. One possible explanation is that at high concentrations, ComEA switches to a different oligomerization conformation than the DNA-bridging form seen at lower concentrations. To explore this hypothesis, we used an optical tweezers DNA overstretching experiment. The setup for this experiment was identical to the force-extension experiment described above, except in this case, the movement of the trap was not stopped when the DNA reached full extension. Instead, the position ramp was continued towards higher forces into the overstretch regime, where applied force breaks down B-DNA and begins to melt it into single-stranded DNA[26,27], before reversing direction. Consistent with previous measurements[26,27], the

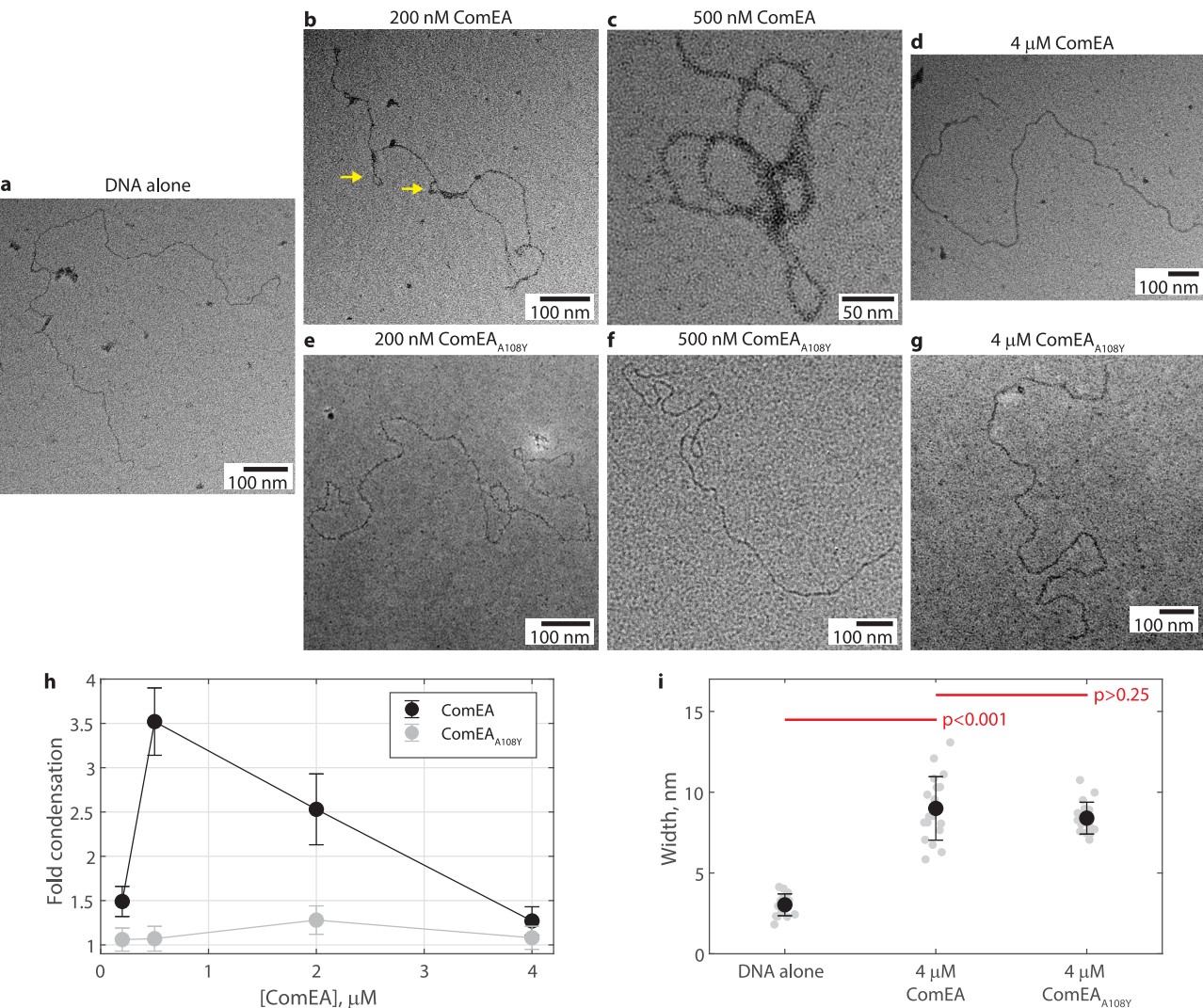

**Fig. 3 | Direct visualization of DNA condensation by ComEA oligomerization.** **a** Representative negative stain electron microscopy image of linearized 5.495 kbp dsDNA. **b**–**d** Representative negative stain electron microscopy images of linearized 5.495 kbp dsDNA treated with the stated concentration of ComEA. Yellow arrows denote loop structures with extended necks. **e**–**g** Representative negative stain electron microscopy images of linearized 5.495 kbp dsDNA treated with the stated concentration of ComEA$_{A108Y}$. **h** The fold reduction in the bounding circle radius of DNA treated with various concentrations of ComEA or ComEA$_{A108Y}$. Data shown as mean ± error propagated from the SEM for $N = 17, 14, 29, 15, 19, 25, 18, 18,$ and 17 DNA molecules for DNA alone, increasing ComEA, and increasing ComEA$_{A108Y}$, respectively. **i** DNA width, with or without 4 µM ComEA or ComEA$_{A108Y}$, measured by negative stain electron microscopy. Individual measurements shown in gray; mean ± SD shown in black for $N = 17, 19,$ and 17 molecules for DNA alone, ComEA, and ComEA$_{A108Y,}$ respectively. Red lines denote the $p$-value from a 2-sample $t$-test (two-sided).

overstretch regime for DNA alone was identified at ~65 pN with reversible horizontal sawtooth patterns on the force-extension curve (Fig. 4a, black). We next repeated this overstretch experiment after injecting a variety of ComEA concentrations. Excitingly, we saw that the amount of force required to overstretch DNA increased with ComEA concentration (Fig. 4a). Less stabilization was observed after injecting ComEA$_{A108Y}$ (Fig. 4b). We determined a degree of stabilization, $\Delta F_{overstretch}$, by measuring the force required for overstretching both before and after injecting ComEA. We found that $\Delta F_{overstretch}$ increased hyperbolically with ComEA concentration, consistent with a simple, saturable binding isotherm (Fig. 4c). The plateau values of the hyperbolic fit revealed that ComEA provided ~16 pN of stability more than ComEA$_{A108Y}$ (Fig. 4c), indicating that oligomerization is present at high ComEA concentrations. The EC50 values of these fits were similar for ComEA and ComEA$_{A108Y}$ ($0.29 \pm 0.29$ µM and $0.21 \pm 0.36$ µM, respectively), and, interestingly, matched the high point of the chevron-shaped plots for fold condensation (Fig. 3h), oligomers per cycle (Fig. S7e), and condensation force (Fig. S7f). Taken together, we

conclude that ComEA forms a different type of oligomer at higher concentrations than at lower concentrations, with the $\Delta F_{overstretch}$ EC50 and chevron-plot high points representing the transition concentration between conformations.

At high ComEA concentrations, we detect oligomerization (Fig. 4c) but not DNA condensation (Fig. 3h), indicating that the connected oligomerization domains must have a great deal of slack such that DNA fluctuations between the DNA binding domains are not restricted. Interestingly, ComEA in diverse Gram-positive bacteria has a largely unstructured linker region between the oligomerization and DNA binding domains[16]. For *Geobacillus stearothermophilus*, this linker (amino acids 125–142, Fig. 1b) is defined as the residues that lacked corresponding electron density in the crystal structure[16]. At higher concentrations, the likelihood of two ComEA molecules landing at adjacent minor groove binding sites along the DNA is greater than at lower concentrations. We thus reason that, at higher concentrations, ComEA switches to a non-bridging oligomer; here the ComEA molecules are bound close enough to each other that the length of their two

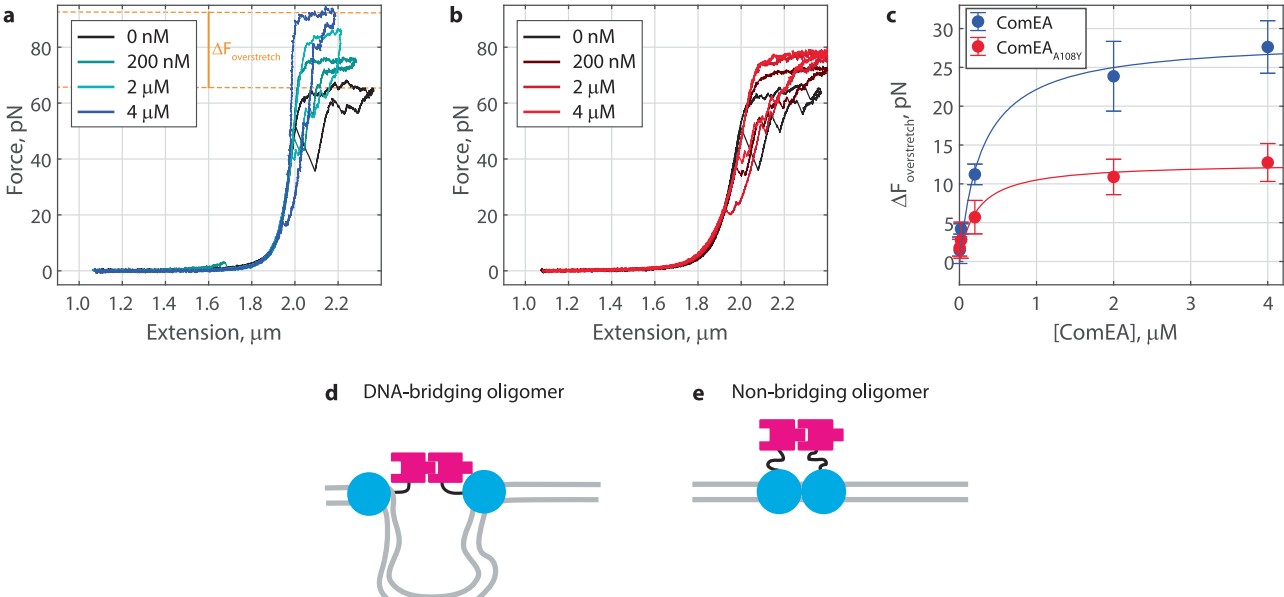

**Fig. 4 | ComEA forms non-bridging oligomers at higher concentrations.**
**a** Example force-extension overstretch curves (pulling and relaxing phases shown) before and after injecting ComEA at the indicated concentrations, highlighting the change in overstretch force, or force where the data appear horizontal. Examples with few bridging oligomers were selected for clarity. The orange dotted lines show measured overstretch forces for buffer alone and 4 µM ComEA. **b** Example force-extension overstretch curves after injecting $ComEA_{A108Y}$ at the indicated concentrations. **c** The increase in the overstretch force as a function of ComEA concentration. Data shown as mean ± SD for $N = 5$, 4, 5, and 4 molecules of DNA for 2 nM, 20 nM, 200 nM, 2 µM, and 4 µM ComEA, respectively, and $N = 5$, 5, 6, 6, and 5

molecules of DNA for 2 nM, 20 nM, 200 nM, 2 µM, and 4 µM $ComEA_{A108Y}$, respectively. Fitted hyperbolas in blue (EC50 = 0.29 ± 0.29 µM and $Y_{max}$=28.6 ± 5.8 pN, fit ± 95% CI) and red (EC50 = 0.21 ± 0.36 µM and $Y_{max}$=12.7 ± 4.4 pN). **d** ComEA in a DNA-bridging dimer. Here, the DNA binding domains (blue) are at distal positions, such that the linker regions (black) get stretched out as the DNA gets stretched out. **e** ComEA in a non-bridging dimer. Here, the DNA binding domains (blue) are at adjacent minor grooves. The linker regions (black) do not get stretched out even when the DNA is at full extension. In this case, overstretching the DNA is required to pull on the linker regions.

---

unstructured linkers combined is greater than the length of the DNA in between them (Fig. 4d, e).

## Tuning the linker length biases ComEA towards forming only bridging or non-bridging oligomers

Our results suggest that the transition point between bridging and non-bridging oligomers is driven by the amount of DNA between the DNA-binding domains of each ComEA molecule, versus the length of the combined linker regions connecting the DNA-binding and oligomerization domains. To test this hypothesis directly, we generated a set of linker mutants. We made a shortened linker mutant, $ComEA_{\Delta L}$, where the disordered central region (amino acids 125–142) was deleted and replaced with a single glycine, and an elongated linker mutant, $ComEA_{LL}$, where 19 amino acids were inserted after residue 143. The 19 residues are identical to the preceding 19 linker residues but in a scrambled order (Fig. 5a). Each mutant eluted as a single, monodisperse peak in size exclusion chromatography, indicating they are each well-folded (Fig. S8). We used a fluorescence polarization assay to measure DNA binding to ensure that these mutants retained the ability to form oligomers. ComEA bound to a 30 bp dsDNA construct, long enough to bind as many as four ComEA molecules, with a $K_D$ of 0.66 ± 0.08 µM and a Hill Coefficient of 1.56 ± 0.28 (Fig. 5b, c). The oligomerization-deficient mutant $ComEA_{A108Y}$ displayed a $K_D$ of 1.91 ± 0.52 µM and a Hill Coefficient within error of unity (0.96 ± 0.16). We thus interpret the positive cooperativity (Hill Coefficient >1) to indicate that ComEA is forming oligomers. $ComEA_{\Delta L}$ and $ComEA_{LL}$ each displayed Hill Coefficients >1, with $K_D$ and $Y_{max}$ values within error of ComEA (Fig. 5b, c). Thus, shortening and elongating the linker region did not interfere with the overall ability of ComEA to form oligomers.

Next, we repeated the optical tweezers force-extension experiment (Figs. 1 and S1) using the ComEA linker mutants. As noted above,

ComEA displayed chevron-shaped behavior for the number of oligomers observed per cycle (Figs. 5d and S7a–e) and for the condensation force generated during the 10-s wait phase (Figs. 5e and S7f), rising from 0 to 150 nM before falling down towards baseline at higher concentrations. As above, this result indicated that DNA-bridging oligomers can form at lower but not higher concentrations of ComEA. In contrast, the $ComEA_{\Delta L}$ mutant did not form observable oligomers or produce condensation force at any concentration tested (Fig. 5d, e), indicating an inability to form DNA-bridging oligomers. Finally, the $ComEA_{LL}$ mutant displayed a shifted chevron pattern, with the peak number of oligomers observed per cycle and the maximum condensation force occurring at 2 µM (Fig. 5d, e). Hence, this mutant continued forming DNA-bridging oligomers even at very high concentrations, unlike ComEA. $ComEA_{LL}$ released more DNA per oligomer than ComEA on average, but still had a prominent peak at 14 bp, consistent with it also lining up at the base of loops (Fig. S9).

Next, we repeated the optical tweezers DNA overstretch experiment to investigate the formation of non-bridging oligomers. We found that ComEA and $ComEA_{\Delta L}$ had high maximum $\Delta F_{overstretch}$ values (28.6 ± 5.8 pN and 31.9 ± 3.8 pN, respectively), whereas $ComEA_{A108Y}$ and $ComEA_{LL}$ had low maximum $\Delta F_{overstretch}$ values (12.7 ± 4.4 pN and 17.9 ± 1.0 pN, respectively; Fig. 5f). Each of the four constructs tested had an EC50 that was within error of each other, indicating that this value is primarily driven by percent occupancy on the DNA. Synthesizing these datasets, we see that ComEA can form both bridging and non-bridging oligomers, $ComEA_{A108Y}$ can form neither bridging nor non-bridging oligomers, $ComEA_{\Delta L}$ is impaired at forming bridging oligomers but can form non-bridging oligomers, and $ComEA_{LL}$ can form bridging oligomers but is impaired at forming non-bridging oligomers. Indeed, our two linker mutants separate function and can each effectively form only one of the two oligomer conformations.

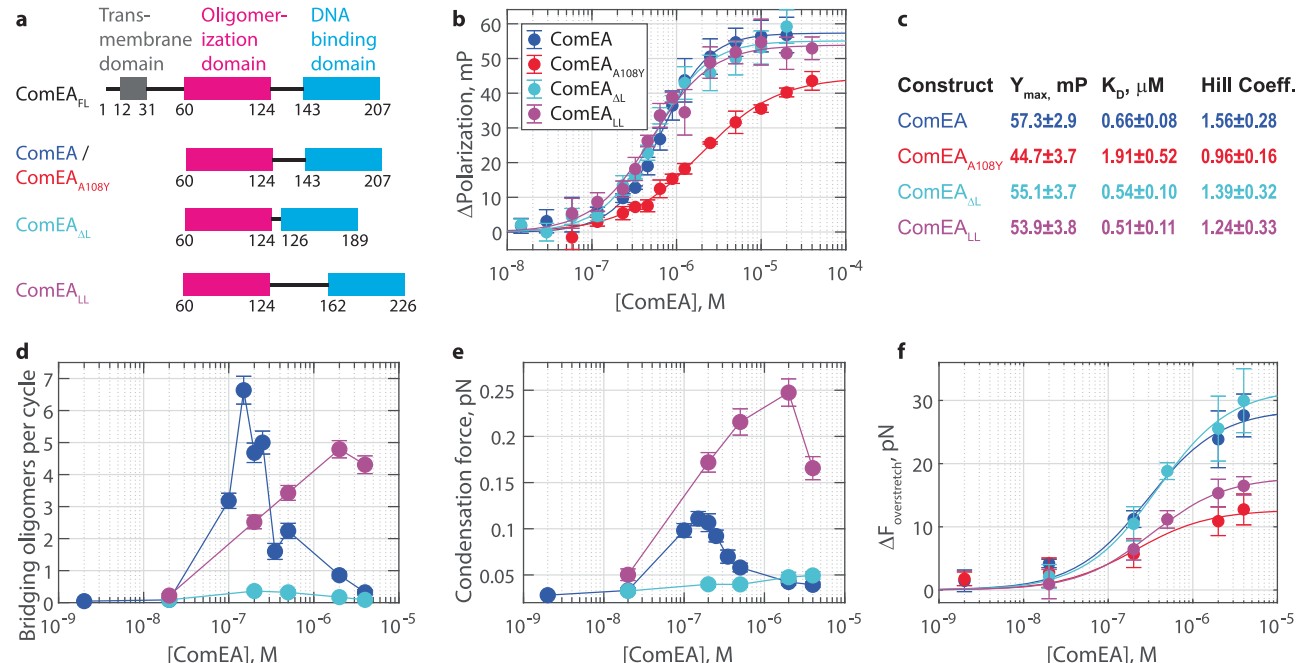

**Fig. 5 | Tuning the linker length biases ComEA towards forming DNA-bridging or non-bridging oligomers. a** Domain diagrams for recombinant ComEA from *Geobacillus stearothermophilus*. **b** Fluorescence polarization measurements of ComEA constructs binding to a 30 bp DNA duplex. Data are shown as mean ± SD for $N = 3$ independent experiments, each run in triplicate. Solid lines show fits to the Hill equation. **c** Fitted coefficients with 95% CI for the Hill equation fits in **b**. **d** The number of DNA-bridging oligomers observed per cycle at various ComEA concentrations. Data plotted as mean ± standard error of the mean for $n = 106$, 105, 100, 96, 144, 94, 75, 100, 98, and 90 cycles off $N = 5, 3, 10, 4, 14, 8, 7, 6, 4$, and 7 molecules of DNA for 2 nM, 20 nM, 100 nM, 150 nM, 200 nM, 250 nM, 350 nM, 500 nM, 2 μM, and 4 μM ComEA, respectively; $n = 80, 94, 87, 87$, and 80 cycles off $N = 6, 8, 9, 10$, and 4 molecules of DNA for 20 nM, 200 nM, 500 nM, 2 μM, and 4 μM ComEA$_{ΔL}$, respectively; and $n = 131, 123, 96, 125$, and 104 cycles off $N = 14, 8, 7, 8$, and 6 molecules of DNA for 20 nM, 200 nM, 500 nM, 2 μM, and 4 μM ComEA$_{LL}$,

respectively. **e** The condensation force generated during the rest phase for each ComEA or mutant concentration. Data plotted as mean ± standard error of the mean, same $n$ and $N$ as **d**. **f** The increase in the overstretch force as a function of ComEA concentration. Data shown as mean ± SD for $N = 5, 4, 5, 5$, and 4 molecules of DNA for 2 nM, 20 nM, 200 nM, 2 μM, and 4 μM ComEA, respectively; $N = 5, 5, 6$, 6, and 5 molecules of DNA for 2 nM, 20 nM, 200 nM, 2 μM, and 4 μM ComEA$_{A108Y}$, respectively; $N = 6, 6, 5, 5$, and 5 molecules of DNA for 20 nM, 200 nM, 500 nM, 2 μM, and 4 μM ComEA$_{ΔL}$, respectively; and $N = 8, 5, 5, 5$, and 5 molecules of DNA for 20 nM, 200 nM, 500 nM, 2 μM, and 4 μM ComEA$_{LL}$. Fitted hyperbolas in blue (EC50 = 0.29 ± 0.29 μM and $Y_{max}$=28.6 ± 5.8 pN, fit ± 95% CI), red (EC50 = 0.21 ± 0.36 μM and $Y_{max}$=12.7 ± 4.4 pN), cyan (EC50 = 0.38 ± 0.18 μM and $Y_{max}$=31.9 ± 3.8 pN), and magenta (EC50 = 0.33 ± 0.07 μM and $Y_{max}$=17.9 ± 1.0 pN) for ComEA, ComEA$_{A108Y}$, ComEA$_{ΔL}$, and ComEA$_{LL}$, respectively.

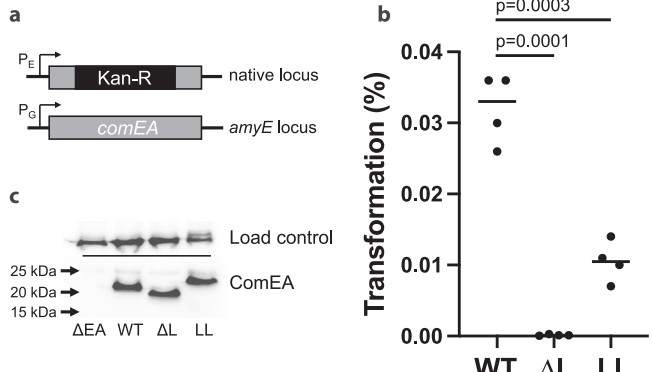

**Fig. 6 | Transformation phenotypes of ComEA short (ΔL) and long (LL) linker mutations. a** ComEA in the native locus was replaced by a kanamycin resistance cassette and complemented by wild-type and mutant *comEA* genes expressed from the competence-dependent *comG* promoter in the ectopic amyE locus. **b** The transformation frequencies supported by wild-type and mutant ComEA were determined in four biological replicates; *p*-value of two-sample *t*-test (two-sided) shown inset. **c** Western blot showing that the wild-type and mutant ComEA proteins were expressed similarly. For a loading control, the top of the gel was cut off (see dotted line) and developed using an antibody raised against elongation factor G. Molecular weight marker positions were determined by alignment with the Ponceau-stained blot, imaged prior to incubation of the membrane with antibody.

## DNA-bridging and non-bridging ComEA oligomers both play a role in natural transformation

To determine the in vivo effects of short- and long-linker mutations, *B. subtilis comEA* coding sequences were expressed under competence control from an ectopic locus and tested for their abilities to complement a *comEA* deletion in trans (Fig. 6a). The short linker mutation removed residues 125 to 139 from ComEA, and the long linker mutation introduced a scrambled version of the same residues just after the wild-type sequence, doubling the linker length. The mean transformation frequencies of the short and long linker strains were, respectively, 220- and 3.2-fold lower than that of the strain expressing wild-type ComEA. Although the short linker phenotype is severe, the mutant protein retains some function because a complete loss of function mutation confers about a 1000-fold transformation deficiency[10]. These results indicate that each of the ComEA oligomer conformations play a role in natural transformation. Force production and DNA condensation during uptake are required, as shown by ComEA$_{ΔL}$. Meanwhile, decondensation is also required, as shown by ComEA$_{LL}$, which can form bridging oligomers and condense DNA. We conclude that condensation and decondensation both play a role in DNA uptake during natural transformation in *B. subtilis*.

## Discussion

In the current work, we find that Gram-positive ComEA forms two different conformations of oligomers on DNA – a DNA-bridging conformation that partially condenses DNA and generates inward pulling

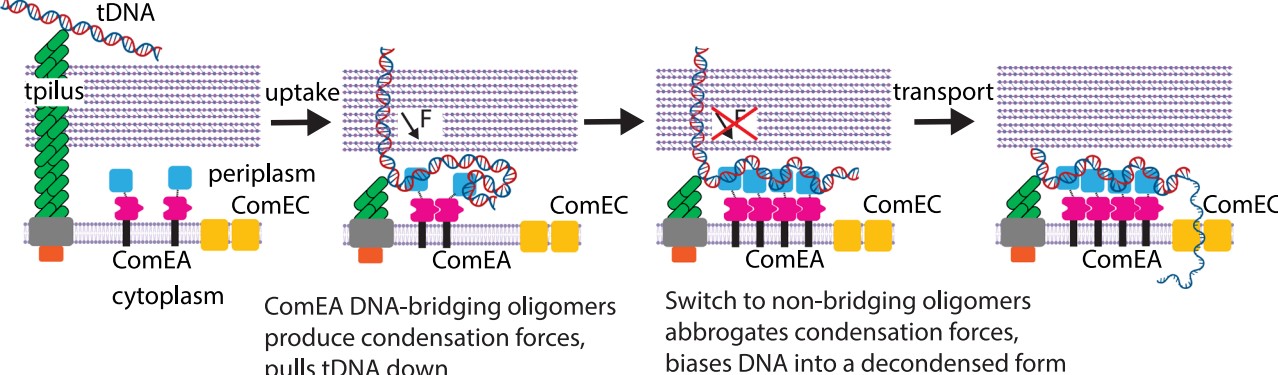

**Fig. 7 | ComEA oligomerization drives DNA uptake by a two-step process.** Diagram showing how ComEA (1) forms DNA-bridging oligomers to produce condensation forces that pull tDNA towards the membrane during DNA uptake, (2) switches to a non-bridging oligomer conformation to turn off force production and decondense DNA during the handoff from DNA uptake into the periplasm to DNA transport into the cytoplasm. Forces generated by ComEA are required to pull additional lengths of tDNA across the cell wall into the periplasm, but must be turned off as they pull in the opposite direction of subsequent DNA transport into the cytoplasm by ComEC. The schematic was created in BioRender. Neiditch, M. (2026) [https://BioRender.com/2c47wtm].

forces of ~0.1 pN, and a non-bridging conformation that decondenses DNA and does not produce force. Switching between the two oligomerization conformations is governed by the local concentration of ComEA oligomerization domains, which can be tuned by either changing the percent occupancy of ComEA on DNA or by changing the length of the unstructured linker region that separates the DNA-binding and oligomerization domains (Fig. 5). Low local concentrations (less ComEA or longer linkers) bias toward the DNA-bridging oligomer conformation, while high local concentrations (more ComEA or shorter linkers) bias toward the non-bridging oligomer conformation. A ComEA mutant that only forms non-bridging oligomers fails to produce condensation forces, while a mutant that only forms DNA-bridging oligomers produces condensation forces under all conditions tested; both lead to impaired transformation efficiency (Fig. 6). Hence, both DNA condensation and decondensation by ComEA are needed for efficient natural transformation in Gram-positive species.

Our data lead to a model where reversible DNA condensation by ComEA drives DNA uptake during natural transformation in a two-step process (Fig. 7). A small loop of transforming DNA is initially brought across the cell wall and into the periplasm by pilus retraction. Subsequently, this tDNA diffuses towards and lands on ComEA embedded in the cell membrane. The ensuing interaction is essential for the irreversible binding of tDNA to the cell[15]. But DNA binding alone is insufficient for uptake, as shown by the ComEA$_{A108Y}$ mutant protein. ComEA rarely oligomerizes in the absence of DNA (>30 μM affinity[16]), meaning molecules in the membrane are distributed and diffusing[28]. Initial interaction of the tDNA with ComEA would most likely be at distal positions, which, after ComEA diffusion and oligomerization would lead to DNA bridges that condense the tDNA and pull it towards the membrane. This pulls additional lengths of tDNA across the cell wall and into the periplasm, which can then be condensed by ComEA similarly and progressively. Over time, more ComEA molecules would diffuse to and accumulate on the DNA, increasing the local concentration of oligomerization domains. The highly dynamic and reversible oligomers (Fig. 2f) then shift towards the non-bridging conformation. This change turns off force production and decondenses the DNA. Reversal of condensation is needed for efficient transformation (Fig. 6b). Indeed, a periplasmic inward pulling force and condensed DNA would be antagonistic towards the subsequent step of DNA transport out of the periplasm and into the cytoplasm (Fig. 7).

ComEA is not an active motor protein and does not have an external energy source. Rather, ComEA oligomerization operates as a passive Brownian ratchet. DNA loops are naturally formed by thermal fluctuations and locked into place by ComEA bridging oligomerization. ComEA cannot form bridging oligomers or produce force if the DNA is held too tautly for loops to naturally form (Fig. S5). The ComEA bridging process is reversible, meaning that a single ComEA pair cannot continuously produce force. However, the collective action of many ComEA pairs can. We refer to ComEA bridging oligomerization as producing force in the narrow definition of its ability to move DNA against an external load[29,30], which we measure here in the sub-pN regime (Fig. 2).

Naturally competent Gram-positive and Gram-negative bacteria have evolved separate mechanisms for DNA uptake[16]. ComEA in Gram-negative species freely diffuses in the periplasm and lacks an oligomerization domain[11–13]. There, ComEA likely serves as a simple Brownian ratchet via DNA binding, disabling backwards diffusion of tDNA after pilus retraction, but doing nothing to pull more tDNA into the periplasm[3,12,14,15]. We previously showed that DNA binding by ComEA without oligomerization is insufficient for natural transformation in *B. subtilis*[16], and here, we show that oligomerization without force production is similarly insufficient (Figs. 5 and 6). Hence, ComEA must produce additional inward pulling force via a second, non-mutually exclusive, oligomerization-based Brownian ratchet to bring tDNA into the periplasm in Gram-positive but not Gram-negative species. We speculate that these differences may arise from the physical differences between cell walls. The relatively thin (<10 nm) peptidoglycan layer plus outer membrane of Gram-negative species may be more permissive to DNA diffusion after pilus retraction. In contrast, the significantly thicker (30–100 nm) peptidoglycan layer of the Gram-positives may provide significantly more friction, such that inward DNA diffusion is retarded[31]. Thus, the additional force generated by ComEA oligomerization is needed.

Why is ComEA a membrane-bound protein that faces outward into the periplasm in Gram-positive species? One possible reason suggested by our data is to keep the force generation directional. If ComEA is anchored on a membrane, any force it generates by stabilizing DNA loops will be inwards towards that membrane. In contrast, if ComEA were free in solution and stabilized a new DNA loop, the DNA on both sides would be pulled inward towards the newly formed oligomer. Several oligomers forming in different positions and pulling against one another would be counterproductive towards pulling DNA into the periplasm. A second reason is that confining ComEA to the membrane may help control the time-resolved transition between DNA-bridging and non-bridging oligomer conformations. Freely

diffusing ComEA would swarm the small lengths of tDNA brought in by pilus retraction, and high-occupancy binding would bias toward non-bridging oligomers, thus precluding the opportunity for force generation. Future work is needed to explore the dynamics of ComEA oligomerization on membranes.

DNA-bridging proteins are common tools for organizing and regulating genetic material. In bacteria, nucleoid-associated proteins such as H-NS (E. coli)[32], Hfq (E. coli)[33], Lsr2 (M. tuberculosis)[34], and Rok (B. subtilis)[35], amongst many others[36], use DNA bridging to compact the genome and to regulate gene expression. Bacterial plasmids and chromosomes are partitioned during cell division by DNA-bridging proteins like ParB[37] and SopB[38]. Proteins like the lac repressor[39] and AraC[40] repressor utilize DNA-bridging versus unbridging to integrate chemical signals and modulate transcription in response. DNA-bridging proteins play more nuanced roles in regulating the topology of the eukaryotic genome—well-known examples include cohesin[41], BAF[42,43], PARP1, and PARP2[44]. ComEA is different from other DNA-bridging proteins in that it does not interact with primary genetic material—its localization to the periplasm necessarily excludes it from playing any direct role in the central dogma. In Gram-positive bacteria, ComEA is membrane-bound, further differentiating it from other DNA-bridging proteins. The reversible condensation of transforming DNA by ComEA instead plays a role in unidirectionally transporting DNA across the cell wall and membrane, to our knowledge, a heretofore undescribed role for a DNA-bridging protein.

A central finding of this work is that groups of *Geobacillus stearothermophilus* ComEA can produce ~0.1 pN of pulling forces when forming bridging oligomers (Fig. 2). Continued condensation occurs against applied loads of ~0.07 pN (Fig. S5). While a condensation force of this magnitude is significant, it is low compared to other DNA-condensing proteins such as HIV-1 nucleocapsid protein (9 pN)[45], spermine (4 pN)[46], ParB from *B. subtilis* (2.1 pN)[47,48], and others measured in the 0.3–1.5 pN regime[43,49–52]. Contributing factors to the weak force production include the lack of DNA bending by ComEA binding and a weak oligomerization on-rate. We argue that the very weak and distinctly partial condensation force generated by ComEA is an evolved feature, as it is significant enough to bias the inward motion of transforming DNA from the extracellular space to the periplasm, but weak enough that it can be easily tamped down by switching oligomerization forms so as not to antagonize the subsequent pulling of DNA from the periplasm to the cytoplasm.

DNA-bridging proteins switching between two conformations is a common trend in biology. Repressors like lac and Arac form bridging dimers to keep genes off, and transfer to non-bridging dimers in response to a chemical ligand, effectively changing DNA topology to turn genes back on[39,40]. H-NS forms bridging versus non-bridging oligomers, possibly to compact nucleoids versus silence genes respectively, and can switch between the two conformations based on divalent cation concentration[53–55]. In comparison, ComEA switches conformation based on local oligomerization domain concentration to turn periplasmic inward pulling forces on and off (Fig. 7). Concentration-based conformational changes are not universal for DNA-bridging proteins: ParB shows saturating condensation force as a function of ParB concentration[47]. ComEA repurposes the same DNA-bridging/non-bridging conformation-switching toolset that is used to turn transcription on and off to instead turn DNA-pulling forces on and off during natural transformation. To our knowledge, ComEA is the first example of a DNA-bridging protein that can repurpose the conformational switching mechanism to transport DNA in this manner.

We propose that ComEA switching to non-bridging oligomers turns off force production to avoid antagonizing the handoff to DNA transport out of the periplasm. Our previous structural data suggest that the non-bridging oligomers may grow into ring- or lock washer-shaped filaments[16]. However, ComEA non-bridging oligomers do not strongly alter the DNA persistence length (Fig. 3h) like H-NS

filaments[54], indicating significant structural differences. Higher resolution structural work is needed to characterize the exact molecular arrangement of non-bridging oligomers. The bridging oligomers we observe here likely include a mixture of bridging and non-bridging dimers. However, even a single bridging dimer within a higher-order oligomer is sufficient to temporarily produce force and thus dominate function. Nonetheless, examining dynamic numbers and conformations of ComEA molecules within each oligomer is of high future interest. The ComEA used for our biochemical studies lacks the transmembrane domain; the effects observed here in free solution likely occur at lower total concentration when restricted to two-dimensional diffusion. However, our in-cell assays (Fig. 6) utilize constructs expressing full-length and membrane-bound ComEA. The reconstitution of reversible DNA condensation by membrane-bound full-length ComEA is of high interest for future studies. Further work is also needed to explore how ComEA is removed from the DNA before or as it is converted to single-stranded DNA for transport. Overall, our work reveals a core mechanism for how DNA is pulled inwards during natural transformation, and more broadly, how conformation-switching DNA-bridging proteins can control DNA topology to transport large molecules across cellular compartments.

## Methods

### Recombinant protein expression and purification

*Geobacillus stearothermophilus* ComEA and ComEA-A108Y were purified as described previously[16]. The ComEA$_{\Delta L}$ and ComEA$_{LL}$ constructs were generated using NEBuilder® HiFi DNA Assembly (New England Biolabs) using the primers mentioned in Table S1. DNA sequencing was used to confirm positive clones. ComEA$_{\Delta L}$ and ComEA$_{LL}$ were purified using the protocol previously described for ComEA and ComEA$_{A108Y}$[16]. ComEA$_{\Delta L}$ and ComEA$_{LL}$ eluted from the Superdex200 10/300 column as a single monodisperse peak.

### Optical tweezers instrument

An optical tweezers instrument following the MiniTweezers[56] design was constructed in advance of this study (see http://tweezerslab.unipr.it/). The dual counterpropagating beam single trap system utilizes two diode lasers (Lumics 376831 and 683568; 808 and 845 nm, 200 and 150 mW) and two water immersion objectives (Olympus UPLSAPO 60XW). The system was calibrated using lambda DNA and has sub-nm spatial resolution at acquisition rates up to 4 kHz. Microfluidic chambers containing three interconnected channels were constructed for these studies, including a micropipette with a <2 μm tip in the middle chamber for holding beads[57].

### Optical tweezers force-extension and overstretch assay setup

The dual-labeled 6 kbp-long dsDNA used for optical tweezers was generated using PCR using a plasmid containing the coding sequence for the very long enzyme Mdn1 as a template[58]. Primers were synthesized with either a 5' biotin group or 5' azide group by Integrated DNA Technologies (Coralville, IA) and purified using a spin column (Qiagen). The resulting 6 kbp product had a biotin and an azide group on either 5' end; its length was verified by gel electrophoresis. To generate DBCO-coated latex beads, 100 μL of 2% w/v 2 μm aliphatic amine beads (Invitrogen A37366) were pelleted by centrifugation, washed, and resuspended in PBS pH = 7.8. DBCO-PEG4-NHS (Broadpharm BP-22288) was then added to a final ~1.5 mM and left to react at room temperature for 1 h. The reaction was quenched with Stop Buffer (20 mM Tris pH = 7.5, 5 mM EDTA), pelleted, washed, and suspended in PBS pH = 7.2 at a final ~0.5% w/v. Streptavidin-coated latex beads were purchased from Spherotech (SVP-20-5). To set up a reaction, 300 ng of dual-labeled DNA was mixed with 0.05% w/v DBCO beads in TL buffer (50 mM HEPES pH 7.5, 55 mM KOAc, 6 mM Mg(OAc)2, and 1 mM DTT) overnight at 4 °C. To form a tether in the optical tweezers

instrument, a streptavidin-coated bead was inserted into the micropipette within the microfluidic and held in place by suction. A DNA-coated DBCO bead was then held in the optical trap and brought into close proximity to fish for a connection. Tethers were verified as single dsDNA strands by the shape of the force extension curve and the overstretch force of ~65 pN in our imaging buffer (20 mM Tris pH = 7.5, 200 mM KCl). ComEA in imaging buffer could then be injected by slowly flowing protein solution through a shunt line as described previously[25,59].

For the force-extension experiments, a script was written to move the optical trap laser back and forth. The DNA was first held at a rest position equivalent to ~55% extension relative to the contour length for 10 s (unless stated otherwise). The trap was then moved forward at 100 nm/s to extend the DNA until 50 pN of tension was reached. The trap was then moved backwards 100 nm/s until the rest position was reached. The breakage of oligomers during pulling is a nonequilibrium process; a fast pull rate was deliberately chosen to resolve all oligomers quickly and at high forces, where changes to the force-extension curve are more pronounced. Approximately 5–50 such cycles were run on a given strand of DNA (spread over several experimental conditions) before the DNA detached from one of the beads either naturally or forcefully by the user. For the overstretch experiments, a similar script was used, except the end position (where the trap stopped moving forward and started moving backward) was selected in real-time by the user once an overstretch was noticed. Data were collected at 200 Hz (instrument response time ~1 ms). All experiments were conducted at $23 \pm 1\,°C$.

### Optical tweezers data analysis
All optical trap data analysis was performed in MATLAB (Natick, MA). The relaxation phase of each force-extension cycle (above 3 pN) was fitted to the extensible worm-like chain equation[17–20]:

$$x(F) = L_0 \left( 1 - \frac{1}{2}\left(\frac{k_B T}{PF}\right)^{1/2} + \frac{F}{S} \right) \quad (2)$$

Using $S = 1200$ pN as a set value and allowing $P$ (constrained to 35–55 nm) and $L_0$ as free parameters. The pulling phase of each cycle was manually observed, and oligomer breakage events were determined as datapoints where the measured force instantaneously dropped while the trap position remained constant. Even sub-pN events were noticeable, but the frequency of oligomers meant that events were most likely undercounted, not overcounted. Force-extension data in-between successive breakage points could then be suitably fitted with the eWLC chain, leaving only $L_0$ as a free parameter.

The amount of DNA released by oligomer breakage was calculating using:

$$DNA\ released = \left(2.94\ \frac{bp}{nm}\right)\frac{\triangle x(F)}{\left(1 - \frac{1}{2}\left(\frac{k_B T}{PF}\right)^{1/2} + \frac{F}{S}\right)} \quad (3)$$

Where $\triangle x(F)$ was determined by comparing the $(x,F)$ datapoint at breakage to the $x$ datapoint on the relaxation phase data with an equivalent $F$ value (average of three local datapoints in each instance). A value of 45 nm was used for $P$ and 1200 pN for S. DNA released per cycle was determined by the first $\triangle x$ measured in the cycle, and the DNA released per oligomer was determined by subtracting successive $\triangle x$ measurements (except for the final $\triangle x$, which required no subtraction).

For measuring condensation forces, the rest phase of each cycle was identified as timepoint where the optical trap was stationary. The data were cleaned using a nonoverlapping 20 data point average-value boxcar filter, effectively downsampling from 200 to 10 Hz. The initial force during rest was determined by taking the average force value

during the 0.5 s recorded. The maximum force generated and held for 1 s was determined using an overlapping 10-data-point average-value boxcar filter and selecting the maximum value.

### DNA amplification for electron microscopy
The 5.495 kbp-long DNA used in these experiments was originally synthesized by GenScript (Piscataway, NJ). This DNA stock was amplified by transforming 100 ng of DNA into 50 μL of competent *Escherichia coli* DH5α cells. The cell-DNA mixture was held at 45 °C for 50 s, then it was pipetted into 1 mL of SOC medium (MP Biomedicals, Santa Ana, CA). The cells were allowed to recover at 37 °C for 1 h. The cells were then centrifuged at 6000 rcf for 1 min at 18 °C, and the supernatant was discarded. The pellet was resuspended in 100 μL of Difco LB Broth (Thermo Fisher Scientific, Waltham, MA), and the cells were streaked onto an LB-agar plate supplemented with gentamycin (7 μg/mL). The plate was incubated at 37 °C overnight, after which, a single colony was selected and transferred into 50 mL of LB supplemented with gentamycin (7 μg/mL) in a sterile 250 mL flask. The cell culture was shaken at 220 rpm at 37 °C overnight. The cells were pelleted by centrifugation at 6000 rcf for 10 min at 4 °C. The cell pellet was separated into 4 tubes and purified using a Qiagen miniprep kit (Hilden, Germany). Cells were resuspended in 250 μL of buffer P1 (50 mM Tris-HCl, pH 8.0; 10 mM EDTA; 100 μg/mL RNAse A) and transferred to a 1.5 mL tube. 250 μL of buffer P2 (200 mM NaOH, 1% (w/v) sodium dodecyl sulfate) was added to the samples, and the samples were inverted 5 times to mix. 350 μL of buffer N3 (Qiagen) was next added to the samples, and they were inverted 5 times to mix, followed by centrifugation at 10,000 rcf for 10 min at 18 °C. 800 μL of each supernatant was added to a Qiaprep spin column (Qiagen) and the columns were centrifuged at ~1000 rcf for 60 s at 18 °C. The flow-through was discarded. 500 μL of buffer PB (Qiagen) was then added to the spin column, followed by centrifugation at 10,000 rcf for 60 s at 18 °C. The flow-through was discarded. Next, 750 μL of buffer PE (10 mM Tris-HCl, pH 7.5; 80% (v/v) ethanol) was added to the spin column, followed by centrifugation at ~13,000 rcf for 60 s at 18 °C. The flow-through was discarded and the spin column was centrifuged at ~10,000 rcf for 60 s at 18 °C one more time. Finally, the spin column was transferred to a clean 1.5 mL microcentrifuge tube. 50 μL of nuclease-free water was added to the column and allowed to stand for 1 min, then the column was centrifuged at 10,000 rcf for 60 s at 18 °C. Purified DNA was present in the flow-through, and its concentration, A260/A280, and A260/A230 were measured with a NanoDrop One$^c$ (Thermo Fisher Scientific).

Miniprep-purified DNA (900 ng/μL) was incubated with BamHI-HF (10 U/μL) (New England BioLabs, Ipswich, MA) in rCutSmart buffer (50 mM potassium acetate; 20 mM tris-acetate; 10 mM magnesium acetate, pH 7.9; 100 μg/mL recombinant albumin) (New England Bio-Labs) at 37 °C overnight. The restriction enzyme was removed using a Qiaprep spin column, as above. The DNA concentration, A260/A280, and A260/A230 were measured again after this step.

### Negative staining electron microscopy and image analysis
For experiments in which we sought to visualize ComEA's ability to condense DNA, the desired concentration of ComEA was incubated with linear 5.495 kbp dsDNA (2.5 ng/μL) for 1 min at 25 °C. A 10 μL droplet of the sample was deposited onto a carbon film, 400 mesh copper grid (CF-400-CU-50) manufactured by Electron Microscopy Sciences (Hatfield, PA) and incubated on the grid for 1 min at 25 °C. The droplet was then wicked off with a piece of filter paper until the grid was dry. 10 μL of pre-dissolved uranyl acetate (1% w/v) stain from Electron Microscopy Sciences was then deposited onto the grid and incubated for 1 min at 25 °C. The droplet was again wicked off with a piece of filter paper until the grid was dry, and the stained grid was loaded into a CM12 transmission electron microscope (Philips, Amsterdam, Netherlands).

DNA strands were typically visible at a minimum of 22,000× magnification. Uncondensed DNA strands were optimally resolved at 28,000–60,000× magnification, and highly condensed DNA strands were optimally resolved at 100,000–120,000× magnification. Live images of the grid were defocused to sharpen the image, and the image's contrast was adjusted after capturing an image.

For an image to be used in our analysis, it must contain the entirety of at least one DNA strand. DNA strands that had portions out of the image's frame or that constituted multiple DNA strands (determined by total DNA length) bound together were considered unsuitable for analysis. A minimum of 14 cumulative images was taken from a minimum of 2 grids for each condition under examination. These images were analyzed with ImageJ (National Institutes of Health; http://imagej.nih.gov/ij/) in two ways. In every image, ImageJ's scale was set to match that of the scale bar. Then, a circle was drawn around the DNA strand such that it was the smallest possible circle that contained the entire strand. The radius of this circle was considered the "size" of the strand. To determine the DNA width, we used the ImageJ line tool at five distinct, selected points along the backbone where the DNA was clearly not overlapping with itself, and then took the average of those five measurements. This process was repeated for every DNA strand included in our analysis. Fold condensation was calculated as the fold difference between the average radius of DNA alone versus the average radius of DNA plus ComEA. For example, images, micrographs were cropped, a smooth filter was applied in ImageJ, and the intensity was rescaled for easy viewing.

### Fluorescence polarization assay for ComEA-DNA binding

The DNA strands used in these experiments were 30-base-pair duplex strands purchased from Integrated DNA Technologies (Coralville, IA) with fluorescein on the 3′ end of one strand. The sequence of our 30 bp duplex was 5′-GCACTGTGAATATCACTACAGCAACCTTGA/36-FAM/−3′ and 3′-CGTGACACTTATAGTGATGTCGTTGGAACT-5′. To perform these experiments, the following were mixed together to 20 μL in a black 384-well microtiter plate (Greiner Bio-One, Kremsmünster, Austria): ComEA (varying concentrations), 30 bp fluorescein-labeled DNA (80 nM), and buffer (20 mM Tris-HCl, pH 7.5; 200 mM KCl). The plate was centrifuged at -15 rcf for 30 s at 20 °C to mix the contents of each well and ensure that no liquid was stuck to the sides of the wells. The plate was loaded into a Spark Multimode Multiplate Reader (Tecan, Männedorf, Switzerland) and fluorescence polarization was measured with fluorescein as the fluorophore. The plate reader was set to use an excitation wavelength of $485 \pm 20$ nm, an emission wavelength of $535 \pm 20$ nm, a G-factor of 1.000, and a gain of 102. Each measurement was made in triplicate as a technical replicate; three independent biological replicates were run and plotted. The data were fit to the Hill equation ($Y = (Y_{max} + x^h)/(K_D^h + x^h)$), where $y$ is the fluorescence polarization, $Y_{max}$ is the theoretical maximum fluorescence polarization value, $x$ is the concentration of ComEA, $h$ is the Hill coefficient, and $K_D$ is the dissociation constant.

### Linker mutation expression constructs and their analysis

For expression in *B. subtilis*, the *comEA* wild-type, ΔL, and LL sequences were expressed from the *comG* promoter at the ectopic *amyE* locus to place them under competence control. For this, the constructs were cloned between the *Eco*RI and *Bam*H1 sites of plasmid pKB149[60]. The fragments were synthesized by Azenta Life Sciences. The ΔL mutation removed 15 residues (125-139) from ComEA. The LL construct added 15 residues after residue 139 in ComEA with the same amino acid composition as the normal linker, but with a scrambled amino acid composition (GGDSQVSGGGGQQVA). The cloned constructs contained a single N-terminal FLAG epitope, but this proved insufficiently sensitive to reliably detect expression. Instead, we performed Western blotting using a rabbit-derived antiserum raised against purified ComEA. The transformation frequencies of the constructed wild-type and mutant

strains were verified by growing them to competence[61] and transforming with genomic DNA, applying selection for leucine prototrophy. The frequency was calculated as (Leu+ transformants/ml)/ (total colony-forming units/ml).

### Reporting summary

Further information on research design is available in the Nature Portfolio Reporting Summary linked to this article.

## Data availability

All data supporting the findings in this paper are provided in the main manuscript and its Supplementary files. Source data are provided within the source data file. Source data are provided with this paper.

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

## Acknowledgements

We thank members of the Mickolajczyk, Neiditch, and Dubnau Labs for useful discussions. We thank Rajesh Patel for his technical knowledge and the RWJMS Core EM facility that he manages. J.S. was supported by Rutgers IRACDA K12 GM093854 from the NIH. Support for this work was provided by the National Institutes of Health through grants R01GM057720 (M.B.N. and D.D.) and R35GM157075 (K.J.M.). Additional support was provided by a Busch Biomedical Grant from the Office for Research at Rutgers (K.J.M.).

## Author contributions

M.B.N., D.D., and K.J.M. conceived of the project. J.S., G.A., and K.J.M. carried out optical tweezer data acquisition and analysis. J.M., H.C., and A.R. carried out fluorescence polarization and electron microscopy experiments. I.A. prepared recombinant protein samples with guidance from M.B. J.H. carried out *B. subtilis* experiments with guidance from

D.D. K.J.M. wrote the paper together with M.B.N. and D.D., and with input from all authors.

## Competing interests

The authors declare no competing interests.
