## [Transparent Peer Review file · Nature Communications]

Reversible DNA condensation drives natural transformation

Corresponding Author: Professor Keith Mickolajczyk

Version 0:

Reviewer comments:

Reviewer #1

(Remarks to the Author)

The presented manuscript describes a reasonable and well supported model of how variable ComEA concentration and structure manipulates DNA and how this interaction may be critical to transformation. The presented experiments appear well performed and analyzed, and the different experimental methods complement one another. However, there are some properties of the fundamental ComEA-DNA interaction that could be further elucidated by additional experiments or analysis. If the authors can revise the manuscript in response to the critiques listed below, this work could be suitable for publication.

The DNA during ComEA incubation is referred to as some fraction extended (e.g. 55%) but it is not justified why these values are given relative to the DNA extension at 50 pN, which is an arbitrary force. A better natural reference point would be the contour length of DNA (DNA is straightened but not elastically stretched), which is quite close but slightly smaller than the 50 pN extension value. Additionally, it is this excess contour length that is not extended between the beads that allows for loop formation, making this a more relevant parameter.

It is frequently stated that Com EA “generates” sub pN condensation forces. However, in the model presented (which seems reasonable), ComEA oligomers do not have a method to actively generate force by pulling disparate regions of DNA together. Rather, ComEA binds transiently formed loops, effectively reducing the length of the template (DNA contour length minus length of DNA sequestered in loops). In an optical tweezer system, in which extension is actively controlled, and template tension is measured as a force on the trapped bead, reduction in effective length while end to end extension is constant will translate into measured force increase. That is, ComEA actively compacts DNA, which translates into a force increase on the tweezers, which is only stable up to ~1 pN. For example, the wormlike chain model, given DNA polymer properties, predicts a DNA held at ~80% of its contour length requires ~0.5 pN of applied force. Thus, it may be helpful to alternatively describe the amount of DNA slack (available DNA template in excess of bead to bead distance) required to allow ComEA bridging. For example, what is the minimum reduction in DNA extension required for bridging? The 77% tested here would seem like the upper limit based on the sub pN forces. Commenting on this relationship would be especially helpful to readers less familiar with force spectroscopy measurement.

The discussion of the most common bridging event size and the presentation of Fig 2C are a bit misleading and seem counterintuitive to the representative force extension curves and electron microscopy images, which show large increases in extension at force drops and large loops of DNA formed respectively. Because of the linear y axis and logarithmic x axis, the high frequency 14 bp dominates the graph and the large bp release events are reduced in size by the scaling. It would be illuminating to alternatively express the amount of DNA sequestered by each size bridging event (multiplying each histogram bin by the DNA released value and a linear x axis). The large events would be much more prominent and help reinforce the point that a small number of oligomers form large loops that compact the DNA while more oligomers bind at the loop stem and contribute minimally to compaction.

Presumably the constant extension experiments reach equilibrium (otherwise continued bridging would raise achieved force), but the force extension curve data is likely far from equilibrium. Given the stated pulling rate 100 nm/s, the representative curves show the formed structures remaining stable up to forces of ~10 pN after ~5 s of force increase. This high stability could be a result of multiple ComEA oligomers bridging the same DNA loop. Ideally, these experiments would be performed at multiple pulling rates to allow extrapolation to equilibrium values of ComEA binding stability in the absence of applied force. The effect of pulling rate should be at least addressed to explain the difference in measured forces. Additionally, the kinetics of bridge formation and disassembly may react differently to varied force. For example, bridge

formation may be completely inhibited by >1 pN force as loops cannot spontaneously form, while bridge disassembly may be less force sensitive but irreversible under increased DNA tension.

Figure 3i shows a two-fold increase in DNA width when saturated with ComEA. However, the method to determine this width should be mentioned in the results, not just the method section. In particular, 5 “random” sections are analyzed, but the thickness clearly changes depending on the type of local structure. The widths of DNA cross overs (perpendicular strands), stabilized bridges (particular DNA strands), and non-self-interacting regions should be measured and averaged separately. Secondly, an alternative measurement method that avoids local fluctuations in width would be to measure the total projected pixel area of each DNA and divide by the known DNA length to calculate an average width without selection bias. Finally, is the scaling of the width and intensity values representative of any fundamental properties of the DNA-ComEA complex? For example, projected volumes of molecular structures measured by atomic force microscopy typically scale with molecular weight, allowing for measurements of protein stoichiometry. Is there any way to estimate the number of ComEA proteins bound to the DNA and compare this with the suspected binding site size, or do the pixel values in electron microscopy not follow any particular scaling trend?

It is shown that ComEA binding does not alter DNA persistence length, but the biophysical significance of this fact is not fully appreciated in the text. More effective condensation agents such as the HIV-1 nucleocapsid protein and spermine mentioned in the discussion locally bend DNA, reducing effective contour length. This allows for the formation of tighter DNA structures such as a DNA toroid which can form with less available DNA slack (less extension reduction) and are stable to higher forces. Thus, if ComEA can only bridge DNA loops that form despite the local ~ 40 nm persistence length stiffness, this help explain why only sub-pN forces are generated/stabilized.

An additional experiment that would be quite useful in examining the fundamental bi-molecular interaction of this system would be concentration change experiments. For example, if bridging oligomers were first formed at low ComEA concentration, then a higher protein was flowed into the sample chamber, would the DNA conformation change? If not, this would indicate the oligomer bridges are quite stable, and if the bridging structures decreased over time (similar to initial incubation with high concentration), this time scale would indicate the temporal stability of these structures. Conversely, if the protein concentration was reduced for a saturated DNA, would bridging events eventually form and at what timescale. Comparison of these rates would additionally measure if the bridging or non-bridging conformation is more stable.

The term condensation is likely not the best term to describe the DNA-ComEA interaction. Generally, DNA condensation proceeds until the entire DNA structure is compacted such as in chromatin formation or during packaging by proteins/polyamines (nucleocapsid/spermine). In addition, for the cases of HIV-1 NC and spermine, the mechanism is completely different, where there are no direct binding interactions between proteins for NC and spermine. The structures observed here are more accurately described as DNA bridging. While it is true that DNA bridging is often an initial step that leads to condensation, here it is shown that ComEA does not proceed to full condensation, and in fact the DNA fully decompacts with increasing ComEA. These structures could be consistently referred to as only “partial condensation”, though DNA bridging is really the best biophysical description, and perhaps DNA compaction could be used as a less specific term.

Reviewer #2

(Remarks to the Author)

Review

Using a combination of optical tweezer force-extension measurements and electron microscopy measurements, the authors demonstrated that ComEA switches between forming DNA condensing oligomers and DNA decondensing oligomers based on concentration. They also elegantly argue how this concentration-dependent switching mode could drive DNA uptake during natural transformation. There is a lot of interesting work here. The article was well-written and a joy to read. There are however a few issues that need to be addressed before suitability of publication can be considered.

Remarks

Page 3:

- “The data were fit well with the extensible worm like chain (eWLC) model (red lines) with parameters matching previously reported values for dsDNA17-20.”

- “In many instances, the force-extension curve between events (blue dotted lines) fit well to the eWLC equation, allowing changes to the contour length but not the persistence length or stretch modulus (Fig. 1d orange curves).”

Why is it assumed here that the persistence length or stretch modulus will not be affected when the DNA is bound by the protein? Would it not make sense for the protein to potentially have a stiffening effect upon DNA binding? Especially considering the stabilizing effect in the overstretching regime addressed later in figure 4. On page 5 it is mentioned that “..., ComEA binds to DNA passively; it does not change the DNA contour or persistence lengths, ...”. The eWLC fit results would be an added benefit here. Also, comparing the force-extension curves in figure S7 there does seem to be some stiffening effect. Also visible from the change in slope in figure 4a with increasing concentration.

Page 5:

- The references to figure 3 seem to have been mixed up.

o “Increasing the amount of ComEA incubated with the DNA from 200 to 500 nM led to more DNA loops and further partial condensation of the DNA (Fig. 3d).” This result is shown in figure 3c.

o “Surprisingly, further increasing the ComEA concentration to 2 or 4 μ M did not lead to even more loops (Fig. 3e, S6).

Instead, the loops disappeared entirely and the DNA decondensed, appearing similar to the condition where no ComEA was present." This result is shown in figure 3d. Additionally, in figure S6 the higher concentrations of ComEA do seem to still show loop formation.

o "DNA loops were not observed with any concentration of ComEAA108Y added, confirming that the loops are formed by ComEA oligomerization (Fig. 3f-g; more examples in Fig. S6)." These results refer to figure 3e-g. However, the DNA in figure 3e appears even more faint than bare DNA shown in figure 3a

Figure 4:

- Figure 4a has an example of DNA with 200nM ComEA barely shows loop formation in comparison to previous examples (figure 2a)

- In Figure 4c $\Delta F > 0$ for $[ComEA] = 0$. How was ΔF here determine?

Figure S1:

- Figure S1b shows a slight increase in extension and force in the second cycle for bare DNA but this is not addressed.

Have you considered doing a velocity sweep for the force-extension measurements? 100nm/s is already considerably fast. It would be interesting to see how the DNA is released when the protein has more time to respond to increasing tension at a slower pulling velocity. I imagine that perhaps the protein can slide first before being released which could lead to smaller loops being pulled. On the flip side, with even faster pulling rates perhaps it shows bigger loops. And how this could change the distribution of DNA released per oligomer shown in figure S3.

Reviewer #3

(Remarks to the Author)

Version 1:

Reviewer comments:

Reviewer #1

(Remarks to the Author)

The authors have adequately addressed our comments and have revised the manuscript accordingly. We recommend publication.

Reviewer #2

(Remarks to the Author)

The response and revision seems complete and suitable.

Reviewer #3

(Remarks to the Author)

REVIEWER COMMENTS

Reviewer #1 (Remarks to the Author):

The presented manuscript describes a reasonable and well supported model of how variable ComEA concentration and structure manipulates DNA and how this interaction may be critical to transformation. The presented experiments appear well performed and analyzed, and the different experimental methods complement one another. However, there are some properties of the fundamental ComEA-DNA interaction that could be further elucidated by additional experiments or analysis. If the authors can revise the manuscript in response to the critiques listed below, this work could be suitable for publication.

The DNA during ComEA incubation is referred to as some fraction extended (e.g. 55%) but it is not justified why these values are given relative to the DNA extension at 50 pN, which is an arbitrary force. A better natural reference point would be the contour length of DNA (DNA is straightened but not elastically stretched), which is quite close but slightly smaller than the 50 pN extension value. Additionally, it is this excess contour length that is not extended between the beads that allows for loop formation, making this a more relevant parameter.

We thank the reviewer for this point and apologize for the confusion. Experimentally, we ran our force-extension curves to a high point of 50 pN before reversing direction. However, all values reported in the paper as “fraction extended” are indeed given relative to the contour length as suggested. To make this clearer, we added text to the main text and methods:

p. 3: “Force extension cycles were then run by (1) holding the DNA at ~55% extension relative to the contour length for 10 seconds, (2) moving the trap forward at 100 nm/s until the DNA was near full extension (50 pN), and (3) moving the trap backwards at 100 nm/s to the resting extension (Fig. S1).”

p. 4: “As an alternate method for measuring ComEA oligomerization forces, we varied how far back from full contour length the DNA was held during the rest phase (Fig. S5a) of each force-extension cycle.”

And to the Material and methods (p. 20):

The DNA was first held at a rest position equivalent to ~55% extension relative to the contour length for 10 seconds (unless stated otherwise).

It is frequently stated that Com EA “generates” sub pN condensation forces. However, in the model presented (which seems reasonable), ComEA oligomers do not have a method to actively generate force by pulling disparate regions of DNA together. Rather, ComEA binds transiently formed loops, effectively reducing the length of the template (DNA contour length minus length of DNA sequestered in loops). In an optical tweezer system, in which extension is actively controlled, and template tension is measured as a force on the trapped bead, reduction in effective length while end to end extension is constant will translate into measured force increase. That is, ComEA actively compacts DNA, which translates into a force increase on the tweezers, which is only stable up to ~1 pN.

We thank the reviewer, and agree completely that more discussion is required to support the term “force production”. ComEA bridging is a Brownian ratchet – it uses chemical bond energy to bias the available thermal fluctuations of DNA to drive the ratchet. The “force generated” more accurately describes the amount of force that the ratchet is capable of opposing. ComEA can and does pull DNA inwards unless that amount of opposing force (~0.1-1 pN) is externally imposed. This is precisely what we measure in the optical tweezers experiments, as you mention above. ComEA is membrane-bound and periplasmic *in vivo*, so this stationary pulling is a suitable reconstitution. ComEA does not actively generate force like an ATPase motor protein does, but it does actively pull DNA inwards even against opposing loads. We found several instances in the literature where other authors refer to this process as force production, although there is some vagueness and debate. Importantly, this type of force production cannot arise from a single molecule (or pair of molecules) like it would from an ATPase motor, because bridge formation is reversible – continued force production requires the collective action of many ComEA pairs, something that we failed to properly describe in the initial submission.

To clarify this point and to make our narrow definition of force production abundantly clear, we have added an additional paragraph to the Discussion section:

“ComEA is not an active motor protein and does not have an external energy source. Rather, ComEA oligomerization operates as a passive Brownian ratchet. DNA loops are naturally formed by thermal fluctuations and locked into place by ComEA bridging oligomerization. ComEA cannot form bridging oligomers or produce force if the DNA is held too tautly for loops to naturally form (Fig. S5). The ComEA bridging process is reversible, meaning that a single ComEA pair cannot continuously produce force. However, the collective action of many ComEA pairs can. We refer to ComEA bridging

oligomerization as producing force in the narrow definition of its ability to move DNA against an external load^{29,30}, which we measure here in the sub-pN regime (Fig. 2).

For example, the wormlike chain model, given DNA polymer properties, predicts a DNA held at ~80% of its contour length requires ~0.5 pN of applied force. Thus, it may be helpful to alternatively describe the amount of DNA slack (available DNA template in excess of bead to bead distance) required to allow ComEA bridging. For example, what is the minimum reduction in DNA extension required for bridging? The 77% tested here would seem like the upper limit based on the sub pN forces. Commenting on this relationship would be especially helpful to readers less familiar with force spectroscopy measurement.

Please see Fig. S5, where we systematically titrate the amount of slack provided. This was our alternative means of determining how much force could be generated. Our answer via this method, 0.07 pN, closely matched the ~0.1 pN measured by holding the trap stationary. We address this description in the new paragraph added to the discussion for the previous comment response.

The discussion of the most common bridging event size and the presentation of Fig 2C are a bit misleading and seem counterintuitive to the representative force extension curves and electron microscopy images, which show large increases in extension at force drops and large loops of DNA formed respectively. Because of the linear y axis and logarithmic x axis, the high frequency 14 bp dominates the graph and the large bp release events are reduced in size by the scaling. It would be illuminating to alternatively express the amount of DNA sequestered by each size bridging event (multiplying each histogram bin by the DNA released value and a linear x axis). The large events would be much more prominent and help reinforce the point that a small number of oligomers form large loops that compact the DNA while more oligomers bind at the loop stem and contribute minimally to compaction.

We thank the reviewer, and we too oscillated on how to best plot this data. The point that you bring up is the point of this plot – the eyes of the viewer are naturally drawn to the dramatic loops (whether in EM or optical tweezers), but those events are quantitatively rare relative to the very small bridging events that reinforce existing loops but do not contribute much to compaction. The amount of DNA sequestered by each bridging event is already what we are measuring (each $\Delta\Delta X$ in Fig.2a for example). We used a logX scale because we indeed see a logarithmic range of values – to plot them on a linear X axis requires a very wide plot with lots of empty white space. We include this linear axis

histogram as **Fig. S3a** and again in comparison to the long linker mutant in **Fig. S9d**. The large loop sizes are very heterogenous and they are not numerically prominent – we did not find that a linear or log X axis changed this.

To improve the manuscript, we added a figure call to the linear X-axis histogram the first time Fig. 2C is mentioned:

p. 4 “The amount of DNA released per oligomer was broadly distributed, with sizes ranging three orders of magnitude, but with a bias towards smaller releases (>50% smaller than 100 bp; Fig. 2c, linear scaling in Fig. S3a).”

Presumably the constant extension experiments reach equilibrium (otherwise continued bridging would raise achieved force), but the force extension curve data is likely far from equilibrium. Given the stated pulling rate 100 nm/s, the representative curves show the formed structures remaining stable up to forces of ~10 pN after ~5 s of force increase. This high stability could be a result of multiple ComEA oligomers bridging the same DNA loop. Ideally, these experiments would be performed at multiple pulling rates to allow extrapolation to equilibrium values of ComEA binding stability in the absence of applied force. The effect of pulling rate should be at least addressed to explain the difference in measured forces. Additionally, the kinetics of bridge formation and disassembly may react differently to varied force. For example, bridge formation may be completely inhibited by >1 pN force as loops cannot spontaneously form, while bridge disassembly may be less force sensitive but irreversible under increased DNA tension.

We thank the reviewer for bringing up this important point. First, we note that the “constant extension” and “force-extension” datasets are two phases of the same experiment. During the “constant extension” phase, we allow for ComEA bridges to form and monitor formation by measuring force. This phase is in equilibrium, as mentioned above and detailed in Fig. 2g-e. Next, we run the “force-extension phase”, during which we have the explicit goal of breaking each oligomer one at a time to measure the number and amount of DNA each oligomer sequestered during the “constant extension” phase. We addressed the kinetics of loop formation and how it depends on force thoroughly and quantitatively in Fig. S5. In this experiment, we varied how much “slack” we provided during the rest phase when ComEA accumulates bridges. Interestingly, ComEA does not form any measurable number of bridges at extensions equating to ~0.07 pN of applied tension on the DNA. Thus, new oligomers are highly unlikely to form during the “force-extension phase”, as we quickly ramp to this force at any pull speed. Indeed, the “force-extension” phase is *not* in equilibrium – we are *only* breaking

oligomers. That said, we are not sure why this is an issue. The goal of the experiment is to resolve each oligomer, which is not an equilibrium process.

“Measured force” at breakage – and more generally the force-dependence of bridge disassembly – is an interesting point, but more complicated than suggested. First, each pulling cycle is unique. There are different ComEA molecules at different positions forming different bridges each time. Said another way, each pull is not independent and identically distributed because the system being pulled on is different each time. This is different than the standard force spectroscopy experiments where exactly one protein or set of proteins is repeatedly pulled upon. Something like the Dudko-Hummer-Szabo model cannot be applied here. Second, the amount of force we measure is the net force felt by the system as a whole, not the amount of force applied to a given oligomer. The applied force may be partitioned over the DNA and one or more bridging oligomers, depending entirely on the spontaneously-formed pattern of oligomer formation during the “constant extension” phase -we cannot know exactly how much force each oligomer feels at any given total measured force. Pulling more slowly does not change this. Lastly, the effective force ramp rate per oligomer cannot be controlled because more than one oligomer can feel force simultaneously – an oligomer can actually experience multiple cycles of increasing and decreasing force in a single pull if several loops were formed (then broken, as force drops after a breakage) and it was the last to break. For the multiple ComEA molecules that we see lined up at the base of loops, it is quite possible that right after the bottommost oligomer breaks, the new bottommost oligomer instantaneously gets loaded with whatever the current force is, regardless of our chosen force ramp rate (which is just how quickly we move the laser). While limiting, we note that the force-dependence of bridge disassembly is not an important metric for the model we present in this paper. The same goes for the equilibrium values of ComEA binding stability in the absence of applied force (a value that would depend strongly on the number of ComEA molecules in the oligomer). We make no claims in the paper that require these values to support them. The equilibrium value of ~ 0.1 pN is more important – membrane-bound ComEA will form bridges and move DNA into the periplasm up until a point where this much tension is felt across the DNA.

We provide here an example of pulling at 10 nm/s to emphasize our points above. Oligomers can break, but do not form. The oligomers break at lower forces since we are pulling more slowly, as suggested. This means that there is a less pronounced change after each break, and less “force-extension” data in between breakages. This leads to ambiguous and difficult-to-analyze data. It is hard to count the number of oligomers in Fig. R1 (is there a breakage at 1.5 pN?); the repeated breakages at ~ 5.5 pN is not relevant to the model in Fig. 7; we already know that multiple ComEA bridges reinforce each loop from Fig. 2c, Fig. 3b, Fig. S3, and Fig. S6., which leads to quick successive

breakage after the first break as force gets redistributed. Pulling slower simply makes the data more difficult to analyze. Conversely, breakages occur faster than we can measure if we increase the ramp rate higher or attempt force clamp experiments. Ultimately, we found little value added to the manuscript by altering the pulling speed.

Figure R1. Example data with 200 nM ComEA on 6 kBP, pulling at 10 nm/s. A force extension cycle was run by waiting at rest (~55% extension relative to contour length) for 10 seconds, pulling forward at 10 nm/s until 50 pN was reached, and then reversing direction at 10 nm/s until the rest position was reached. Breakage events at low forces are more difficult to detect.

To improve the manuscript, we added the following statements on bridge formation kinetics and the effect of pulling speed on measured force at oligomer breakage events:

p. 5: “Altogether, these results show that DNA-bridging ComEA oligomers can condense DNA until tension in the 0.1-1 pN range is reached, at which point new bridges fail to form and condensation stalls.”

p. 20: “The breakage of oligomers during pulling is a nonequilibrium process; a fast pull rate was deliberately chosen to resolve all oligomers quickly and at high forces, where changes to the force-extension curve are more pronounced.”

Figure 3i shows a two-fold increase in DNA width when saturated with ComEA. However, the method to determine this width should be mentioned in the results, not just the method section. In particular, 5 “random” sections are analyzed, but the thickness clearly changes depending on the type of local structure. The widths of DNA cross overs (perpendicular strands), stabilized bridges (particular DNA strands), and non-self-interacting regions should be measured and averaged separately. Secondly, an alternative measurement method that avoids local fluctuations in width would be to measure the total projected pixel area of each DNA and divide by the known DNA length to calculate an average width without selection bias. Finally, is the scaling of the width and intensity values representative of any fundamental properties of the DNA-ComEA complex? For example, projected volumes of molecular structures measured by atomic force microscopy typically scale with molecular weight, allowing for measurements of protein stoichiometry. Is there any way to estimate the number of ComEA proteins bound to the DNA and compare this with the suspected binding site size, or do the pixel values in electron microscopy not follow any particular scaling trend?

We thank the reviewer for the opportunity to improve our description of how we measured DNA width. It is important to note that this data was obtained using negative stain electron microscopy – what we visualize is the stain, not the DNA or ComEA. Getting perfectly uniform stain is not easy, and using an area method as suggested would not work because overlapping regions of DNA do not necessarily get twice as much stain and thus appear twice as dark. Neither the darkness of the stain nor the width of the DNA is equivalent to the height measurement in atomic force microscopy. There is not a scaling trend in either measurement that allows us to estimate stoichiometry – we would need Cryo-EM or a higher resolution method to determine this. The width does, however, tell us that the object getting stained is wider in diameter than bare DNA, and we know from Fig. 5b that binding is fully saturated. This is sufficient to justify our claims that ComEA is bound to the DNA at 4 μ M despite the lack of DNA loop structures.

In response, we have improved our methods section and added a description in the main text and methods as suggested.

p. 5: “We confirmed that ComEA was bound to the DNA at high concentrations by measuring the DNA thickness at nonoverlapping regions; incubation with 4 μ M ComEA or ComEAA108Y led to DNA that was significantly thicker than the no protein added condition, but not from each other (Fig. 3i).”

p. 22: “To determine the DNA width, we used the ImageJ line tool at five distinct, selected points along the backbone where the DNA was clearly not overlapping with itself, and then took the average of those five measurements.”

It is shown that ComEA binding does not alter DNA persistence length, but the biophysical significance of this fact is not fully appreciated in the text. More effective condensation agents such as the HIV-1 nucleocapsid protein and spermine mentioned in the discussion locally bend DNA, reducing effective contour length. This allows for the formation of tighter DNA structures such as a DNA toroid which can form with less available DNA slack (less extension reduction) and are stable to higher forces. Thus, if ComEA can only bridge DNA loops that form despite the local ~40 nm persistence length stiffness, this help explain why only sub-pN forces are generated/stabilized.

We appreciate the opportunity to discuss this point. While DNA binding (without oligomerization) does not change the persistence length, the formation of DNA bridges does indeed change the local shape of DNA by locking in very rare bending conformations. We see very high local curvatures in our EM images of ComEA (200-500 nM) on DNA (Fig. 3). Building on this point, we do see many bridges that are much smaller than the persistence length, some even smaller than 10 bp (Fig. 2c). This could, theoretically, open up even tighter DNA structures to be formed by thermal fluctuations as mentioned. However, condensation is still very limited.

We would like to re-emphasize Fig. S5, where we titrate the amount of slack provided. We see very close to zero bridging oligomers form per cycle on average when the amount of slack provided is ~25% (1.5 kbp of slack in our 6kbp DNA). 1.5 kbp should be capable of transiently forming loop structures by thermal fluctuations, even if ComEA binding doesn't change the persistence length – the amount of applied force here is only ~0.1 pN. But no oligomers form. In all cases, ComEA oligomers continue to form up until 0.1 pN of tension is across the DNA. The force-dependent limitation of oligomer formation is more likely due to the oligomerization on-rate than due to the persistence length of DNA. However, as you mention, it may help achieve higher forces if ComEA binding were to bend DNA.

In response, we added text to the Discussion:

(p. 9) “Contributing factors to the weak force production include the lack of DNA bending by ComEA binding and a weak oligomerization on-rate”.

An additional experiment that would be quite useful in examining the fundamental bi-molecular interaction of this system would be concentration change experiments. For example, if bridging oligomers where first formed at low ComEA concentration, then a higher protein was flowed into the sample chamber, would the DNA conformation

change? If not, this would indicate the oligomer bridges are quite stable, and if the bridging structures decreased over time (similar to initial incubation with high concentration), this time scale would indicate the temporal stability of these structures. Conversely, if the protein concentration was reduced for a saturated DNA, would bridging events eventually form and at what timescale. Comparison of these rates would additionally measure if the bridging or non-bridging conformation is more stable.

We agree, and attempted the experiments suggested here numerous times both before and after initial submission. However, we ran into technical difficulties that prevented us from obtaining satisfactory data.

We inject ComEA into our microchamber through a shunt line (pulled capillary for microinjection) as slowly as possible ($<1 \mu\text{L/s}$), but even slight injection flows produce a force on the beads in our experiments. The amount of force from injecting is greater than the 0.1 pN coming from ComEA, meaning that we cannot feasibly monitor de-condensation by measuring force at a constant trap position while injecting ComEA. When we wait for the injection flow to slow down ($\sim 10+$ seconds later), the changeover in ComEA conformation is already complete. The dynamics are faster than we can capture given the experimental limitations.

On the other hand, we had difficulty removing protein from our chamber (reducing the concentration from $2 \mu\text{M}$ to 200 nM) without compromising the experiments. This is because clearing the chamber requires direct flow through the channel, as opposed to addition from protein, which can be done slowly through a shunt line. The high forces from the main channel flow typically destabilize our experiments, and the beads often fly out of the optical trap.

To improve our manuscript in response to this comment, we added the following text to page 5:

“Sequentially injecting higher concentrations of ComEA onto a single stand of DNA (e.g. $4 \mu\text{M}$ after 150 nM) led to changeover in behavior to the higher concentration faster than could be experimentally measured.”

The term condensation is likely not the best term to describe the DNA-ComEA interaction. Generally, DNA condensation proceeds until the entire DNA structure is compacted such as in chromatin formation or during packaging by proteins/polyamines (nucleocapsid/spermine). In addition, for the cases of HIV-1 NC and spermine, the mechanism is completely different, where there are no direct binding interactions between

proteins for NC and spermine. The structures observed here are more accurately described as DNA bridging. While it is true that DNA bridging is often an initial step that leads to condensation, here it is shown that ComEA does not proceed to full condensation, and in fact the DNA fully decompacts with increasing ComEA. These structures could be consistently referred to as only “partial condensation”, though DNA bridging is really the best biophysical description, and perhaps DNA compaction could be used as a less specific term.

We appreciate this point, and emphasize that we qualify the DNA condensation process by ComEA as “partial” many times throughout the text:

p. 3 subsection title: ComEA-driven DNA condensation is partial

p. 4: “Taken together, our data indicate that as ComEA forms more and more bridging oligomers on DNA, it tends to reinforce loops that have already been sequestered rather than sequestering new loops, leading to only partially condensed DNA.”

p. 5: Increasing the amount of ComEA incubated with the DNA from 200 to 500 nM led to more DNA loops and further partial condensation of the DNA (Fig. 3d).

p. 5: “This limited degree of condensation is consistent with the limited DNA released per cycle in the optical tweezers experiment (Fig. 2b).”

The process is partial whether we use the word “condensation” or “compaction”. ComEA cannot fully condense or fully compact DNA. We believe that the “partial” qualifier is much more important than the term condensation versus compaction. Thus, to improve the manuscript, we have added further remarks in the discussion on the partial nature of DNA condensation by ComEA:

p.7: “In the current work, we find that Gram-positive ComEA forms two different conformations of oligomers on DNA – a DNA-bridging conformation that partially condenses DNA and generates inward pulling forces of ~0.1 pN, and a non-bridging conformation that decondenses DNA and does not produce force.”

p. 9: “While a condensation force of this magnitude is significant, it is low compared to other DNA-condensing proteins such as HIV-1 nucleocapsid protein (9 pN)⁴³, spermine (4 pN)⁴⁴, ParB from *B. subtilis* (2.1 pN)^{45,46}, and others measured in the 0.3-1.5 pN regime^{41,47-50}. We argue that the very weak and distinctly partial condensation force generated by ComEA is an evolved feature, as it is significant enough to bias the inward motion of transforming DNA from the extracellular space to the periplasm, but weak

enough that it can be easily tamped down by switching oligomerization forms so as not to antagonize the subsequent pulling of DNA from the periplasm to the cytoplasm.”

Reviewer #2 (Remarks to the Author):

Review

Using a combination of optical tweezer force-extension measurements and electron microscopy measurements, the authors demonstrated that ComEA switches between forming DNA condensing oligomers and DNA decondensing oligomers based on concentration. They also elegantly argue how this concentration-dependent switching mode could drive DNA uptake during natural transformation. There is a lot of interesting work here. The article was well-written and a joy to read. There are however a few issues that need to be addressed before suitability of publication can be considered.

Remarks

Page 3:

- “The data were fit well with the extensible worm line like chain (eWLC) model (red lines) with parameters matching previously reported values for dsDNA17-20.”

- “In many instances, the force-extension curve between events (blue dotted lines) fit well to the eWLC equation, allowing changes to the contour length but not the persistence length or stretch modulus (Fig. 1d orange curves).”

Why is it assumed here that the persistence length or stretch modulus will not be affected when the DNA is bound by the protein? Would it not make sense for the protein to potentially have a stiffening effect upon DNA binding? Especially considering the stabilizing effect in the overstretching regime addressed later in figure 4.

We thank the reviewer for this point. Please note that we are referring here to regions in between oligomer breakage events during a given force extension curve (between blue dotted lines) at low [ComEA]. During these segments, we are pulling out the DNA in between bridging oligomers – which should behave like bare DNA until force starts to get placed onto a bridging oligomer. For this reason, we only allowed the contour length to change to see if the data could be well fit. The ability for us to fit eWLC depends on how many datapoints we get in between oligomer breaking events, which is outside our control. Additionally, many segments did not follow eWLC behavior, possibly due to sliding or very fast successive breaks of higher order oligomers (Fig. 1d purple). The point of this analysis was to show, as an example, that fitting eWLC to “regions in between” was only successful in certain instances, and not a universal method for determining how much DNA was sequestered by ComEA bridging oligomers. For this

reason, we used the “ ΔX ” method shown in Fig. 2 to determine the amount of DNA sequestered by the loop (as opposed to comparing eWLC fits).

To improve the manuscript, we altered our text to declare more clearly that it is a simplifying assumption:

p. 3: “In many instances, the force-extension curve between events (blue dotted lines) fit well to the eWLC equation under the simple assumption that only the contour length could change (Fig. 1d orange curves).”

And we emphasize our text where we declare that fitting eWLC to these regions is not a universally useful method:

p. 3: “Although the subsections of the curve between oligomer breaking events could not be fit to the eWLC model in every instance, most commonly due to the subsection being very short, we could determine the datapoint before each breakage and compare it to a point of equivalent force on the subsequent relaxation curve (Fig. 2a).”

Finally, please see our response directly below to the potential stiffening of DNA by binding at higher ComEA concentrations - where non-bridge oligomers occur and a net effect of the entire force-extension curve is feasible.

On page 5 it is mentioned that “..., ComEA binds to DNA passively; it does not change the DNA contour or persistence lengths, ...”. The eWLC fit results would be an added benefit here. Also, comparing the force-extension curves in figure S7 there does seem to be some stiffening effect. Also visible from the change in slope in figure 4a with increasing concentration.

We thank the reviewer for the opportunity to go into this point with more detail. Addition of ComEA at high concentrations leads to an apparent stiffening in the force-extension curves, but we believe that this is mainly due to the change in stretch modulus (in turn, associated with the very large measured changes to overstretch force). When we fit the 4 μM ComEA data (where ComEA is bound but rarely in bridging oligomers), we find a persistence length of 62.1 ± 16.7 nm and a stretch modulus of 4.45 ± 4.79 nN. The persistence length seems high, but is technically within error of our DNA alone measurements. The measured value for the stretch modulus is *very* high – which makes sense because ComEA binding and non-bridging oligomerization each increase the overstretch force. A higher stretch modulus and a longer persistence length each lead to “stiffer” looking FE curves. Overall, the errors in these fits are quite high – likely due to

the option of “stiffening” the curve by changing either parameter – and we do not feel like they add value to the manuscript.

We do, however, have a different way of looking at differences in persistence length in the absence of applied force, where changes to a stretch modulus will have little impact on the data. We took negative stain EM images of dsDNA with and without 4 μM ComEA WT and A108Y. In both cases, ComEA coated the DNA (Fig. 3i) but did not lead to a significant increase in the measured radius of the DNA molecule (as expected for an increase in persistence length), nor did it lead to any apparent straightening along the backbone. This evidence supports the claim that ComEA non-bridging oligomers have a weak effect on thermally-driven DNA bending fluctuations (changing the persistence length).

To improve the manuscript, we added the following text to the EM section (p. 5):

“This result indicates that ComEA binds to DNA at high concentrations, but does not strongly impact thermally-driven fluctuations of DNA bending.”

We also softened our statement quoted by the reviewer above on ComEA not changing the persistence length, since it is not critical to our argument and “stiffening under load” versus “change to persistence length” may be confusing to readers (p. 6):

“Our optical tweezers and electron microscopy data suggest that at high concentration, ComEA binds to DNA but does not induce condensation.”

Finally, we tempered our statement on changes to persistence length in the discussion (p. 9):

“However, ComEA non-bridging oligomers do not strongly alter the DNA persistence length (Fig. 3h) like H-NS filaments, indicating significant structural differences.”

Page 5:

- The references to figure 3 seem to have been mixed up.

o “Increasing the amount of ComEA incubated with the DNA from 200 to 500 nM led to more DNA loops and further partial condensation of the DNA (Fig. 3d).” This result is shown in figure 3c.

We thank the reviewer for finding this mistake. It has been corrected in the revised manuscript.

o “Surprisingly, further increasing the ComEA concentration to 2 or 4 μM did not lead to even more loops (Fig. 3e, S6). Instead, the loops disappeared entirely and the DNA decondensed, appearing similar to the condition where no ComEA was present.” This result is shown in figure 3d. Additionally, in figure S6 the higher concentrations of ComEA do seem to still show loop formation.

We fixed the text to call figure 3d. Loops are less common at higher concentrations, as shown in the “chevron plot” in Fig. 3h. There are still some loops at 2 μM , as our examples show, but fewer than at 500 nM. To improve the manuscript, we have revised the text to

“Surprisingly, further increasing the ComEA concentration to 2 or 4 μM did not lead to even more loops (Fig. 3e, S6). Instead, the loops progressively disappeared and the DNA decondensed, appearing similar to the condition where no ComEA was present.”

o “DNA loops were not observed with any concentration of ComEAA108Y added, confirming that the loops are formed by ComEA oligomerization (Fig. 3f-g; more examples in Fig. S6).” These results refer to figure 3e-g. However, the DNA in figure 3e appears even more faint than bare DNA shown in figure 3a

We appreciate this point, and note that the darkness/faintness of the DNA has to do with the application of stain, not with the formation of loops. In response, we have gathered new data and updated Fig. 3e as well as Fig S6 with example images that have improved staining. These improved examples make it more clear that ComEA_A108Y does not induce loop formation.

Figure 4:

- Figure 4a has an example of DNA with 200nM ComEA barely shows loop formation in comparison to previous examples (figure 2a)

We picked this example on purpose for clarity in plotting. The breakage of loops overwhelms the plot visually, such that it is not easy to compare the curves at different concentrations. In response, we have updated the Fig. 4a caption to include the text: “Examples with few bridging oligomers were selected for clarity.”

- In Figure 4c $\Delta F > 0$ for $[\text{ComEA}] = 0$. How was ΔF here determine?

We thank the reviewer and note that the data shown in Fig. 4C is not for $[\text{ComEA}] = 0$, but rather for 2 nM, 20 nM, and upwards. In response, we have updated the Fig. 4c caption to read:

“The increase in the overstretch force as a function of ComEA concentration. Data shown as mean \pm SD for N=4-6 molecules of DNA at each concentration (2 nM to 4 μM).”

Figure S1:

- Figure S1b shows a slight increase in extension and force in the second cycle for bare DNA but this is not addressed.

We thank the reviewer for this point. The slight increase noted is due to missed reads from the detector. Note that the increase is in both channels (extension and force) and does not appear when force is plotted against extension. In response, we have updated the fig S1b caption to read;

" Data showing the extension of 6 kbp DNA over time following the protocol in panel a. Shown below is the resulting force over time. The blip in the second cycle is due to missed time points from the detector."

Have you considered doing a velocity sweep for the force-extension measurements? 100nm/s is already considerably fast. It would be interesting to see how the DNA is released when the protein has more time to respond to increasing tension at a slower pulling velocity. I imagine that perhaps the protein can slide first before being released which could lead to smaller loops being pulled. On the flip side, with even faster pulling rates perhaps it shows bigger loops. And how this could change the distribution of DNA released per oligomer shown in figure S3.

We appreciate this point – and tried pulling at different velocities as part of this revision. We found that pulling slower simply led to more ambiguous results – ComEA most likely slid more, but bridges also broke at lower forces which made data analysis substantially harder. An example is in Fig. R1 above. In contrast, when we pulled faster, the oligomers broke very quickly and it became difficult to count each one. We do, however, know that “bigger loops” cannot be seen because our two methods of measuring force (by measuring at constant extension, Fig. 2, and by measuring the amount of DNA released during the “force-extension” phase as a function of DNA slack provided in the “constant extension” phase, Fig. S5) are in quantitative agreement. If substantial sliding

was occurring prior to the first breakage event, then the force estimated in Fig. S5 would be smaller – because it is determined from “DNA released” while the direct constant position measurement is determined from reduction in length (DNA sequestered). Ultimately, however, we did not find any significant new insights from running velocity sweep experiments.

Reviewer #3 (Remarks to the Author):

REVIEWERS' COMMENTS

Reviewer #1 (Remarks to the Author):

The authors have adequately addressed our comments and have revised the manuscript accordingly. We recommend publication.

Thank you!

Reviewer #2 (Remarks to the Author):

The response and revision seems complete and suitable.

Thank you!

Reviewer #3 (Remarks to the Author):

Thank you!